# 2D Titanium carbide printed flexible ultra-wideband monopole antenna for wireless communications

Weiwei Zhao[1,5], Hao Ni[2,5], Chengbo Ding[1], Leilei Liu[2] ✉, Qingfeng Fu[2], Feifei Lin[1], Feng Tian[3], Pin Yang[1], Shujuan Liu[1], Wenjun He[1], Xiaoming Wang[1], Wei Huang ®[1,4] ✉ & Qiang Zhao ®[1,2] ✉

Flexible titanium carbide ($Ti_3C_2$) antenna offers a breakthrough in the penetration of information communications for the spread of Internet of Things (IoT) applications. Current configurations are constrained to multi-layer complicated designs due to the limited conformal integration of the dielectric substrate and additive-free $Ti_3C_2$ inks. Here, we report the flexible ultrawideband $Ti_3C_2$ monopole antenna by combining strategies of interfacial modification and advanced extrusion printing technology. The polydopamine, as molecular glue nano-binder, contributes the tight adhesion interactions between $Ti_3C_2$ film and commercial circuit boards for high spatial uniformity and mechanical flexibility. The bandwidth and center frequency of $Ti_3C_2$ antenna can be well maintained and the gain differences fluctuate within ±0.2 dBi at the low frequency range after the bent antenna returns to the flat state, which conquers the traditional inelastic Cu antenna. It also achieves the demo instance for the fluent and stable real-time wireless transmission in bending states.

The advancement of the Internet of Things (IoT) system greatly demands the seamless integration of radio-frequency (RF) antennas and circuits at wide frequency band for device-device communication[1,2]. Ultrathin and flexible antenna components have arisen a great interest towards reliable wireless connectivity with miniaturized and wearable electronics, including sensors, displays, data processing devices, etc[3]. The key materials are expected to be flexible with high electrical conductivity and tolerable mechanical deformation[4]. In comparison with conventionally used metals[5], carbon-based nanomaterials[6], and polymers[7], two-dimensional titanium carbide ($Ti_3C_2$) is becoming nominated star materials for RF antennas due to its intrinsic high electrical conductivity (10,000-20,000 S cm$^{-1}$), good skin depth (2.4 GHz, 10 μm), excellent mechanical strength and easy processibility[8–11]. For example, in 2016, Gogotsi et al. first explored the potential applications of $Ti_3C_2$ nanomaterials in the flexible dipole antennas in the WiFi frequency band (2.4 GHz)[8]. In their later work, a breakthrough shows that 5.5 μm-thick $Ti_3C_2$ patch antenna has a comparable radiation efficiency (>99%) at 16.4 GHz, which is almost comparable with that of a standard 35 μm-thick copper patch antenna. It makes $Ti_3C_2$ promising for integrated RF communications in flexible and wearable IoT devices. However, two major challenges remain on the construction of $Ti_3C_2$ antenna. On the

[1]State Key Laboratory of Organic Electronics and Information Displays & Jiangsu Key Laboratory for Biosensors, Institute of Advanced Materials (IAM), Nanjing University of Posts & Telecommunications, 9 Wenyuan, Nanjing 210023, P. R. China. [2]College of Electronic and Optical Engineering & College of Flexible Electronics (Future Technology), National and Local Joint Engineering Laboratory of RF Integration and Micro-Assembly Technology, Nanjing University of Posts & Telecommunications, 9 Wenyuan, Nanjing 210023, P. R. China. [3]Key Lab of Broadband Wireless Communication and Sensor Network Technology, Nanjing University of Posts and Telecommunications, 9 Wenyuan, Nanjing 210023, P. R. China. [4]Frontiers Science Center for Flexible Electronics (FSCFE), MIIT Key Laboratory of Flexible Electronics (KLoFE), Northwestern Polytechnical University, Xi'an 710072, P. R. China. [5]These authors contributed equally: Weiwei Zhao, Hao Ni. ✉e-mail: liull@njupt.edu.cn; provost@nwpu.edu.cn; iamqzhao@njupt.edu.cn

one hand, the utilization of polyethylene terephthalate (PET) and double-sided tape between $Ti_3C_2$ layer and commercial circuit boards causes the sophisticated manufacturing process as well as hinders the direct conformal integration with flexible electronics and chips, thus leading to unsatisfactory power delivery and sensitive resonant frequency in wireless communication[12]. On the other hand, the working bandwidth is relatively narrow, and it is difficult to meet the ultra-wideband requirements. Hence, it is necessary to exploit flexible ultrawideband $Ti_3C_2$ monopole antennas fabricated through the progressive microfabrication technique.

As one of the representative direct ink printing protocols, the extrusion printing technique has been a revolutionary and eco-friendly manufacturing route for the mass production of flexible integrated electronics with high-resolution geometry patterns and digital customization[13–15]. It not only can generally deposit the functional viscoelastic inks with a large concentration window and suitable fluidic properties (e.g., surface tension and viscosity) under ambient conditions[16–18], but also has apparent advantages in realizing high-precision conformal printing on different substrates (whether flat or curved) without additional masks and accessories, as well as avoiding time-consuming and complicated transfer processes, which is superior to previously reported screen printing, physical vapor deposition, and spray coating, etc[2,19–21]. Substantial progresses demonstrate that additive-free $Ti_3C_2$ inks have inherent properties, including excellent dispersion quality, negative surface charge, and hydrophilicity[13]. Accordingly, they are proved to be particularly suitable for printing electronics, including transparent electrodes[22], transistors[23], photodetectors[24], energy storage devices[25], and sensors[26]. It provides a paradigm for the construction of patch antennas with highly compact

and intricately shaped components. However, as a core component, the commercial dielectric substrate is short of interfacial adhesion with active functional materials caused by chemically inert, smooth, and hydrophobic surface[6]. Thus, it is also highly desirable to combine the interface optimization process and extrusion printing technology for manufacturing flexible ultrawideband $Ti_3C_2$ monopole antennas capable of compact integration.

Here, we report the direct extrusion printing technology of additive-free concentrated $Ti_3C_2$ inks for flexible ultrawideband $Ti_3C_2$ monopole antenna. The polydopamine (PDA) is chosen as a molecular glue nano-binder between $Ti_3C_2$ film and dielectric substrate, contributing to the conformal integrated microstrip transmission lines (TLs) and antennas with high spatial uniformity and mechanical flexibility. The reflection coefficient $S_{11}$ and gain of $Ti_3C_2$ antenna can be well maintained after cyclic bending. The demonstration of wireless movie transmission in bending states is well achieved through the wireless communication platform.

## Results
### Extrusion printing for flexible ultrawideband $Ti_3C_2$ monopole antenna

The multi-layered (m-) $Ti_3C_2$ is synthesized by selective removal of the aluminum layer from the commercial $Ti_3AlC_2$ phase (Fig. 1a and Supplementary Fig. 1). Delaminated $Ti_3C_2$ is prepared through bath sonication. The abundant negative electrostatic charges (e.g., -F, -OH, and -O) on the hydrophilic $Ti_3C_2$ nanosheets lead to stable aqueous inks (Supplementary Fig. 1c). The concentrated viscous $Ti_3C_2$ inks can be directly extrusion-printed for flexible patterned patch antenna (Fig. 1b and Supplementary Fig. 2). The printing technique is also

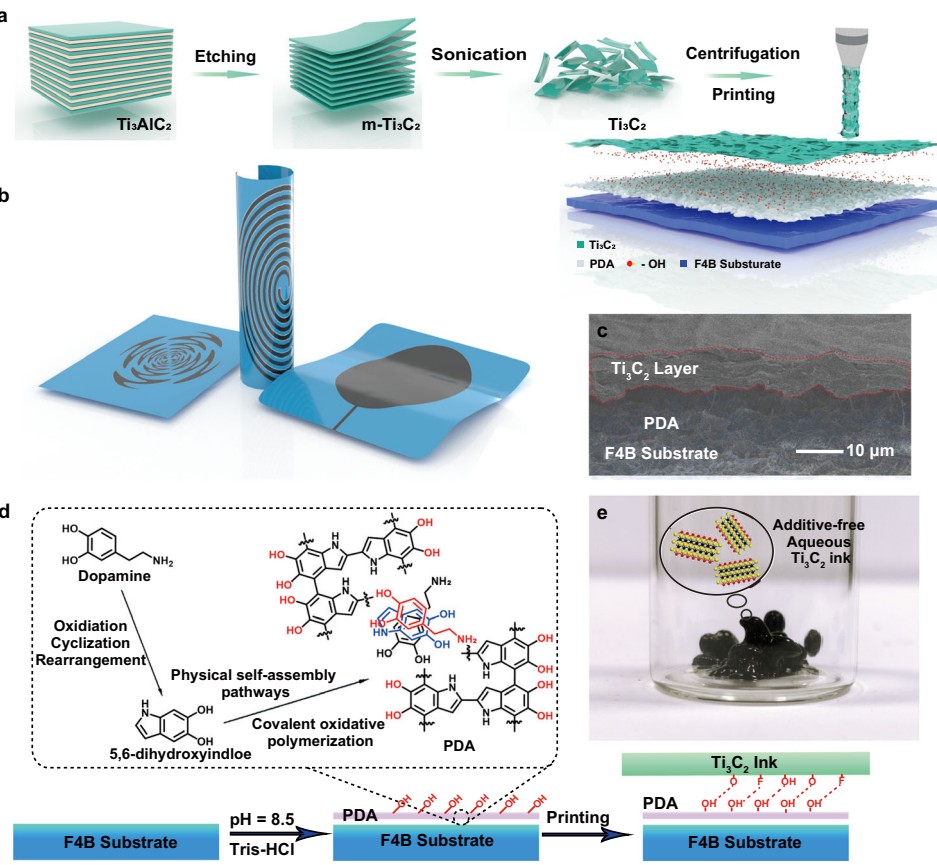

**Fig. 1 | Preparation of $Ti_3C_2$ for antenna. a** The schematic pathways for ultrathin $Ti_3C_2$ nanosheets and the direct extrusion printing of $Ti_3C_2$ inks for flexible antennas. **b** Different patterns of flexible $Ti_3C_2$ antennas. **c** The cross-sectional SEM image of $Ti_3C_2$ antennas including $Ti_3C_2$ layer, polydopamine (PDA) and F4B220M (F4B) substrate. **d** Schematic illustration of preparation process for $Ti_3C_2$ antennas. **e** The optical photograph of $Ti_3C_2$ inks.

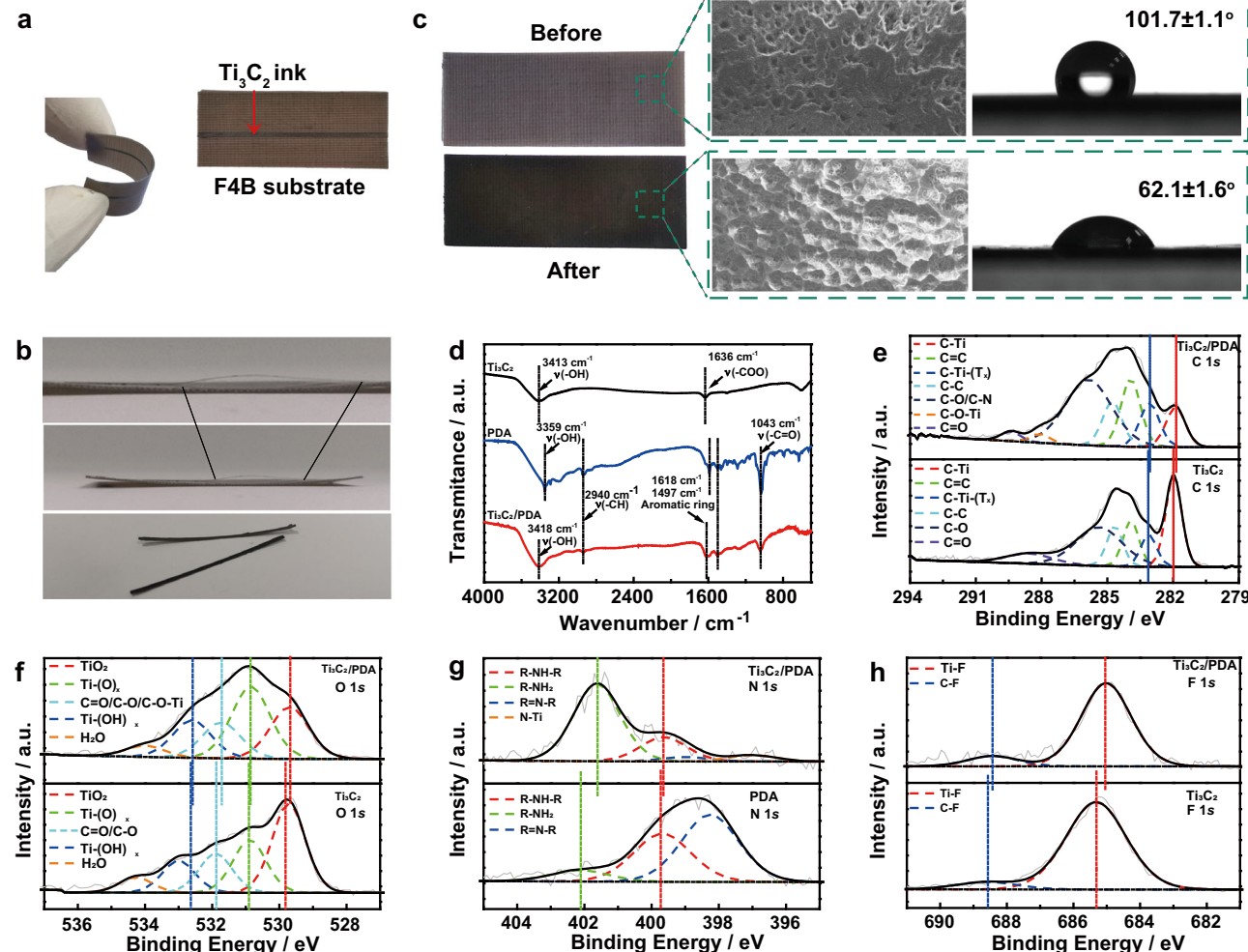

**Fig. 2 | Characterization of Ti₃C₂ microstrip transmission lines (TLs). a** Digital images of flat and bent Ti₃C₂ microstrip TLs with PDA coating. **b** Digital images of Ti₃C₂ microstrip TLs without PDA coating and the exfoliated Ti₃C₂ layer. **c** Digital image, SEM image, and contact angle of F4B dielectric substrate before and after PDA depositing. **d** FT-IR spectra of Ti₃C₂, PDA and Ti₃C₂/PDA. **e** XPS C 1*s*, **f**, XPS O 1*s*, **g** XPS N 1*s*, and **h** XPS F 1*s* of Ti₃C₂/PDA, Ti₃C₂ or PDA.

precise for complicated flexible electronic circuits (Fig. 1b and Supplementary Fig. 3). The interconnected multilayer Ti₃C₂ film can be uniformly coated on PDA-modified commercial polytetrafluoroethylene dielectric substrate (The model is F4B220M. It is abbreviated as F4B substrate in this work) (Fig. 1c and Supplementary Fig. 4). Concretely, the F4B dielectric substrate is first modified with dopamine (DA) monomers in Tris-HCl buffer solution (pH = 8.5) (Fig. 1d). Then, the self-polymerization of DA monomers occurs via the intramolecular cyclization and intermolecular polymerization[27]. The formed PDA layer acts as a secondary platform to improve the interfacial adhesion interactions between the dielectric substrate and printed Ti₃C₂ film, contributing the conformal integrated Ti₃C₂ antennas with the high spatial uniformity and mechanical flexibility (Fig. 1e).

**Flexible and compact Ti₃C₂ microstrip TLs**

The rational design of TLs is vital for low transmission loss to minimize signal attenuation and distortion[28]. To achieve the compact integration, Ti₃C₂ TL is directly extrusion-printed on the PDA-modified F4B dielectric substrate (Fig. 2a). The total thickness of different layers is lower than that of previous reports (Supplementary Table 1). The as-fabricated Ti₃C₂ microstrip TLs can be bent at a large bending angle, showing excellent mechanical stability (Fig. 2a). While, the Ti₃C₂ TL is easy to fall off from the F4B dielectric substrate in the absence of PDA coatings (Fig. 2b). It shows that PDA irreplaceably enables to enhance

the adhesion strength between Ti₃C₂ layer and F4B dielectric substrate. The surface properties of F4B dielectric substrate are systematically analyzed before and after DA treatment (Fig. 2c). Scanning electron microscopy (SEM) images show that the surface roughness of F4B dielectric substrate increases after the uniform adhesion of PDA coatings. The contact angle of water on the F4B dielectric substrate decreases from 101.7 ± 1.1° to 62.1 ± 1.6°[29]. It proves that the hydrophilicity is improved due to the exposure of the hydroxyl functional groups from PDA[30]. Fourier transform infrared (FT-IR) spectrum of Ti₃C₂/PDA is composed of the main characteristic peaks of Ti₃C₂ and PDA (Fig. 2d). The peaks at around 1618 and 1497 cm⁻¹ show obvious shifts in comparison with those of PDA and Ti₃C₂, suggesting the interaction between catechols/quinone groups in PDA and terminal groups (-OH/-O/-F) of Ti₃C₂[31]. X-ray photoelectron spectroscopy (XPS) of Ti₃C₂/PDA demonstrates the surface compositions of C, N, Ti, O, and F elements at around 285.0, 401.2, 456.5, 531.2, and 685.3 eV, respectively (Supplementary Fig. 5a). Compared with pristine Ti₃C₂, Ti₃C₂/PDA has a newly appeared peak at 288.0 eV in C 1*s* spectrum, which is assigned to the catechol-titanium coordination bond (C-O-Ti) (Fig. 2e)[32]. In addition, C-Ti and C-Ti-(T)ₓ peaks downshift towards the lower binding energy due to the electron transfer from PDA to Ti₃C₂[33]. For O 1*s* spectrum of Ti₃C₂/PDA, the quinone state (C = O) peak possibly overlaps with C-O-Ti peak (Fig. 2f)[34,35]. The downshift tendency for Ti-(OH)ₓ, Ti-(O)ₓ, and TiO₂ is consistent with C 1*s*[36]. Ti 2p spectrum also has the similar changes (Supplementary Fig. 5b). Compared with PDA,

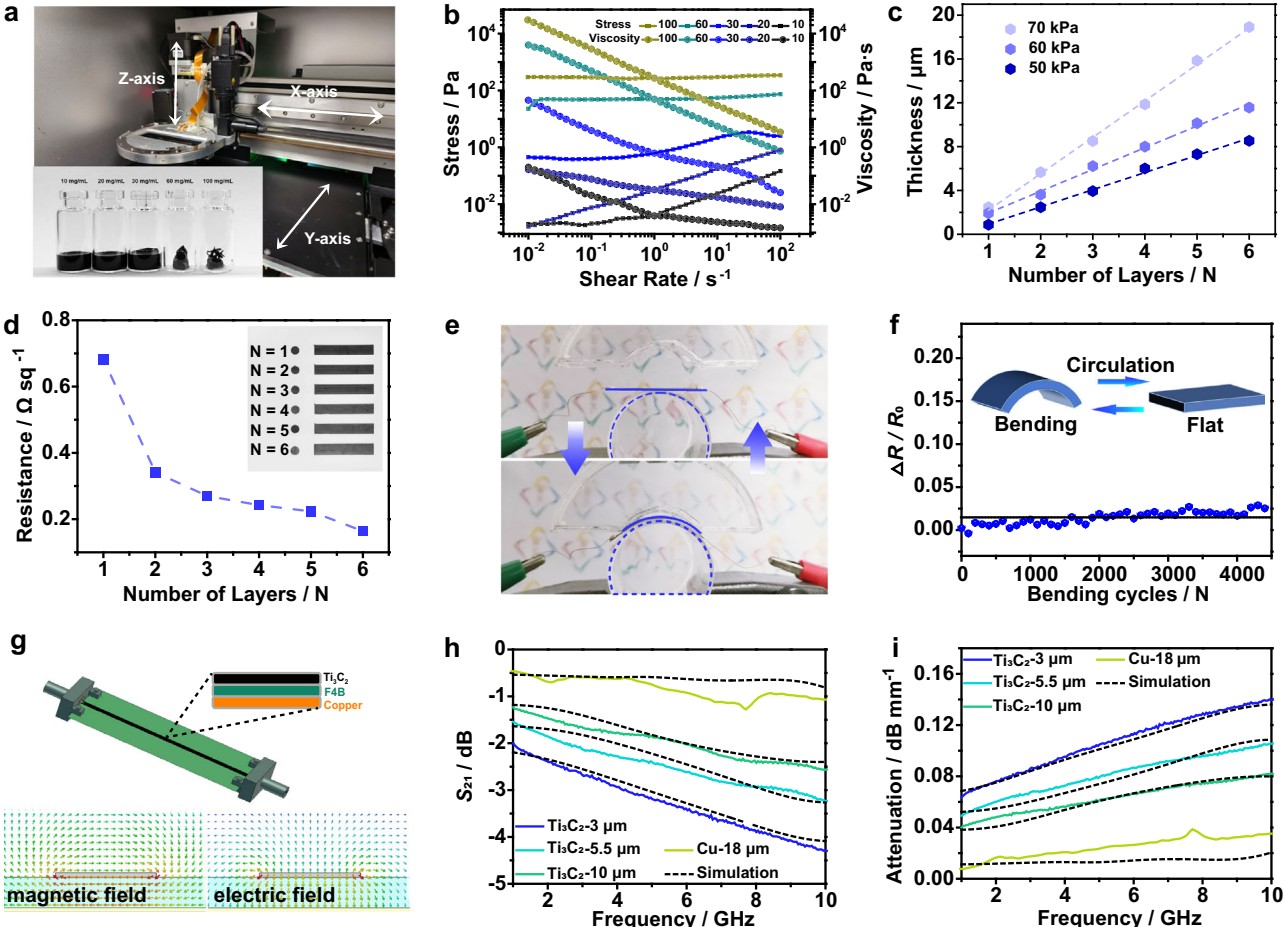

**Fig. 3 | Extrusion printing for Ti₃C₂ microstrip TLs. a** The internal structure of a microelectronic printer. Inset: Photographs of various Ti₃C₂ inks. **b** Stress and viscosity plotted as a function of shear rate. **c** Thickness plotted as a function of layer number. **d** Sheet resistance plotted as a function of layer number. Inset: Optical images of various printed lines (3 cm in length, 50 kPa) with different layer numbers. **e** Diagram of the bending test. **f** $\Delta R/R_0$ at different bending cycles. **g** Schematic configuration of Ti₃C₂ microstrip TLs. Inset: The cross-sectional view shows the materials in different layers. Simulated electromagnetic field distribution of Ti₃C₂ TLs. Transmission coefficient $S_{21}$ (**h**) and attenuation constant (**i**) of 3 cm-long Ti₃C₂ TLs with different thicknesses.

the lower binding energy of the primary amine (R-NH-R) and secondary amine (R = N-R) peaks is ascribed to the hydrogen bonding interactions between PDA and Ti₃C₂ (Fig. 2g)[37]. The new N-Ti peak at 396.8 eV is assigned to the binding of amine at the unterminated Ti sites. The binding energy downshifts of F 1*s* (i.e., Ti-F and C-F) are mainly attributed to the hydrogen bonding with -O groups of PDA (Fig. 2h). The binding affinity and hydrogen bonding between PDA and Ti₃C₂ are the main connection modes[37]. Therefore, PDA can act as the role of molecular glue nano-binder to reinforce the compact interaction between F4B dielectric substrate and Ti₃C₂ layer. The PDA treatment provides a universal and simple strategy to connect Ti₃C₂ materials and dielectric substrate for microstrip TLs and antennas.

**Performance measurement of Ti₃C₂ microstrip TLs**

In the extrusion printing, the solid-like ink is extruded as a filament through a nozzle and deposited on the substrate for a fine-resolution printing[38,39]. The printing pass can be well adjusted by the accurate orientation of X-axis and Y-axis (Fig. 3a and Supplementary Fig. 6). The extrusive strength is controlled by atmospheric pressure. The printing height is adjusted along the Z-axis. For the Ti₃C₂ inks with different concentrations of 10, 20, 30, 60, and 100 mg mL⁻¹, the ink's viscosity increases with the raised concentrations at a constant shear rate and decreases with the increase of shear rates at a certain of concentration (Fig. 3b). It presents a clear shear-thinning behavior and non-Newtonian fluid characteristics for continuous extrusion printing

process[40]. While, the stress has the contrary tendency to viscosity. In combination with the viscoelastic curves and Hershel-Bulkley fluid model, the optimal ink concentration is selected as 60 mg mL⁻¹ with the yield stress of 48 Pa, which is the most suitable parameter for the actual extrusion printing (Supplementary Figs. 7 and 8)[41]. By adjusting the extrusive pressure (i.e., 50, 60, and 70 kPa), the thickness measured by step profiler can be effectively adjusted (Fig. 3c and Supplementary Fig. 9). For example, the thickness gradually increases from 8.4 μm for 50 kPa to 12.1 μm for 60 kPa and 14.1 μm for 70 kPa at the layer number (N) of 6. The thickness linearly increases with the layer number at a certain pressure of 50 kPa. That is 1.2, 2.7, 3.8, 5.9, 7.7, and 8.4 μm for the layer number of 1, 2, 3, 4, 5, and 6, respectively. The sheet resistance of Ti₃C₂ film significantly decreases from 0.68 Ω sq⁻¹ to 0.16 Ω sq⁻¹ with 6-layer overprints (Fig. 3d). As the 3 μm-thick Ti₃C₂ TLs are periodically bent at a curvature radius of 1.5 cm (Fig. 3e), the relative resistance only has <2% change after 5,000 bending cycles, showing the excellent cyclic bending stability of Ti₃C₂ TLs (Fig. 3f).

Ti₃C₂ microstrip TLs can work through directly connecting with two pressed connectors without soldering (Fig. 3g). The simulated electromagnetic field distribution of quasi-TEM mode for Ti₃C₂ microstrip TLs agrees with that made of conventional metal[21]. Ti₃C₂ microstrip TLs with three different thicknesses (i.e., 3, 5.5, and 10 μm) are prepared and named as Ti₃C₂-3 μm, Ti₃C₂-5.5 μm and Ti₃C₂-10 μm, respectively. Scattering parameters (S-parameter $S_{11}$ and $S_{21}$) of all Ti₃C₂ microstrip TLs are measured at the frequency range from 1 to

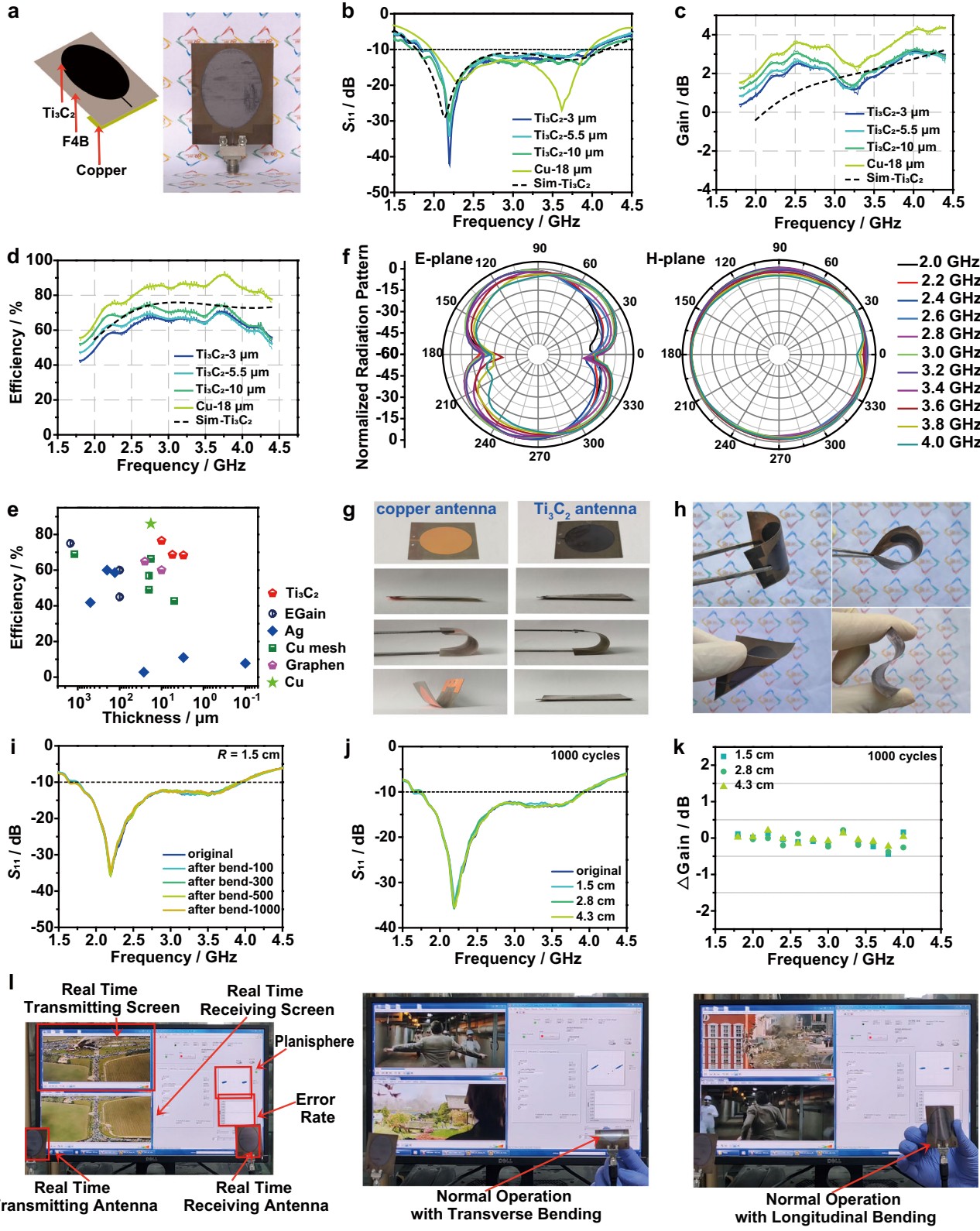

**Fig. 4 | Characterization and application for ultrawideband Ti₃C₂ monopole antennas. a** Schematic, and optical photograph of flexible Ti₃C₂ antennas on dielectric substrate with a pressed connector. **b** Measured and simulated $S_{11}$ parameter of Ti₃C₂ antennas. Measured and simulated gain (**c**) and radiation efficiency (**d**) of Ti₃C₂ antennas. **e** A comparison of radiation efficiency versus thickness for Ti₃C₂ with metal and other materials as patch antennas. **f** Typical radiation pattern of Ti₃C₂−5.5 μm antennas measured in the anechoic chamber. The unit is dBi. **g** Digital photographs of copper antennas and Ti₃C₂ antennas. **h** Digital photographs of Ti₃C₂ antennas in various deformations. $S_{11}$ parameter of Ti₃C₂−5.5 μm antennas after different bending cycles at $R = 1.5$ cm (**i**) and after 1000 bending cycles under different bending radii (**j**). **k** Gain difference of Ti₃C₂−5.5 μm antennas after 1000 bending cycles under different bending radii. **l** Demonstration of Ti₃C₂ antennas for wireless communication.

10 GHz using a vector network analyzer (Fig. 3h and Supplementary Fig. 10). The commercial 18 μm-thick copper TLs with the same geometry are manufactured as a reference. $S_{11}$ of $Ti_3C_2$ microstrip TLs is kept below −10 dB (Supplementary Fig. 10). Transmission coefficient $S_{21}$ of all TLs decreases with increasing frequencies in the range of 1-10 GHz due to the skin depth effects[42]. $S_{21}$ also increases with the increased thickness of $Ti_3C_2$ microstrip TLs because of the decreased sheet resistance (Fig. 3c, d). The attenuation constant ($\alpha$) of $Ti_3C_2$ microstrip TLs is calculated by $S_{11}$ and $S_{21}$ (Fig. 3i). It increases with frequency for a certain thickness and decreases with thickness from 3 to 10 μm. It is worth noting that the attenuation constant of $Ti_3C_2$−10 μm microstrip TL is only 0.102 dB mm$^{-1}$ at 2.4 GHz while that of the copper TL is 0.052 dB mm$^{-1}$. Even for $Ti_3C_2$−3 μm TL, the attenuation constant only increases to 0.171 dB mm$^{-1}$. Although the attenuation constant is worse than that of 18 μm-thick copper TL, it is still acceptable to be used as a substitute for traditional metal in RF communication systems[8,43]. The measurement of the $Ti_3C_2$ antenna in the later part supports this point.

## Ultrawideband $Ti_3C_2$ monopole antenna

The simulated and measured models of the ultrawideband $Ti_3C_2$ monopole antennas are designed with $Ti_3C_2$ patch, dielectric substrate, and ground plane (Fig. 4a and Supplementary Fig. 11-12). During the test, the coaxial cable is connected through the pressure contact without soldering. Three $Ti_3C_2$ antennas with different thicknesses (i.e., 3, 5.5, and 10 μm), which are named as $Ti_3C_2$−3 μm, $Ti_3C_2$−5.5 μm and $Ti_3C_2$−10 μm, respectively, are fabricated for measurements (Supplementary Fig. 13 and Supplementary Table 2). A copper antenna with the same design is used for comparison. The reflection coefficient $S_{11}$ is lower than −10 dB in working frequency band of 1.7-4.0 GHz, indicating that the antenna can well receive the energy input by the vector network analyzer's feeder (Fig. 4b). It is well matched with the simulation results. As the thickness of $Ti_3C_2$ components decreases, the deeper notch of $S_{11}$ curves appears due to stronger local resonance characteristics and decreased radiation efficiency of the antenna[44,45]. The relative bandwidth of the $Ti_3C_2$ antennas reaches 75% ± 3%, covering WLAN, Bluetooth, 5 G (n41, n78) frequency bands and far exceeding other $Ti_3C_2$ antennas at this stage[8,21,43]. Its long-term stability lays the foundation for practical applications (Supplementary Fig. 14). The simulated current distribution of $Ti_3C_2$ monopole patches is identical to that of a metallic monopole antenna (Supplementary Fig. 15). The loss of $Ti_3C_2$ antenna is slightly higher than that of Cu antenna in high-frequency region, which makes $Ti_3C_2$ antennas have the lower $Q$ value. The working bandwidth of $Ti_3C_2$ antennas at −10 dB is expanded and is comparable to the traditional Cu antenna[22]. The measured gain and total efficiency of $Ti_3C_2$ antenna are in the same trend as the copper antenna, which increases with the thickness due to the decreased conductor loss (Fig. 4c, d). The gain of $Ti_3C_2$ antenna is about 1 dBi less than that of the copper antenna, but remains at a high gain for a monopole antenna overall. It can be further increased through improving the conductivity of $Ti_3C_2$ component, using alternative substrate with lower dielectric loss, increasing antenna orientation, or combining multiple antennas into an antenna array[2,46]. The radiation efficiency reaches 68.4% for $Ti_3C_2$−3 μm, 68.7% for $Ti_3C_2$−5.5 μm, and 76.5% for $Ti_3C_2$−10 μm, which may be further increased through improving the conductivity of $Ti_3C_2$ layer or adopting suitable substrates with lower dielectric loss[2,46]. The increased gain and efficiency tendency with the thickness of $Ti_3C_2$ patches are attributed to the decreased conductor loss[47]. The radiation efficiency at 2.4 GHz of $Ti_3C_2$ antenna is only 14% ± 5% lower than the copper counterpart. It is also higher than antennas made of other materials (Fig. 4e and Supplementary Table 3). The normalized radiation patterns of $Ti_3C_2$ antenna are consistent with the pattern of the standard monopole antenna (Fig. 4f). It is a 8 shape on E-plane and a circle on H-plane. The three-dimensional (3D) radiation patterns

obtained by the full-wave simulations software intuitively show that the $Ti_3C_2$ monopole antenna can radiate omnidirectionally on H-plane (Supplementary Figs. 16-18). The as-fabricated $Ti_3C_2$ antenna can be randomly twisted, showing excellent flexibility (Fig. 4g, h and Supplementary Movie 1). For comparison, the copper monopole antenna with the same structure has no resilience at any bending angles. The absence of PDA adhesive layer between $Ti_3C_2$ and F4B dielectric substrate results in structural instability (Supplementary Fig. 19). The cyclic bending tests are further performed to investigate the stability of $Ti_3C_2$ antenna. Typically, the $Ti_3C_2$−5.5 μm antenna is first bent for 100, 300, 500, and 1000 cycles at the bending radius ($R$) of 1.5 cm (Fig. 4i). There is no obvious shift for bandwidth and center frequency after the antenna returns to the flat state. $S_{11}$ values also have negligible changes after 1000 bending cycles at the bending radii of 1.5, 2.8, and 4.3 cm (Fig. 4j). The corresponding gain differences slightly fluctuate within ±0.2 dBi at the low-frequency range (Fig. 4k), which is comparable to the previously reported work[21]. The performance can also be well maintained in the bent state with the bending radii of 1.5, 2.8, and 4.3 cm (Supplementary Fig. 20). The $Ti_3C_2$ antennas with the thickness of 3 and 10 μm also have similar results (Supplementary Figs. 21-24). Thus, the excellent cyclic bending stability of $Ti_3C_2$ antenna is well demonstrated for application in flexible RF devices. The actual communication performance of the $Ti_3C_2$ antenna is conducted using NI USPR-2943R platform. Two $Ti_3C_2$ antennas are connected to two antenna ports of the wireless communication platform through coaxial cables as transmitting and receiving antennas, respectively (Fig. 4l). The entire test process is real-time, using Binary Phase Shift Keying (BPSK) modulation technology. The upper left part of the computer monitor is the real-time transmission screen, the lower left part is the real-time receiving screen, and the right half is the communication status information, including the operating frequency, the distribution diagram of the signal vector endpoints (Planisphere) and the bit error rate diagram. The transmission performance of $Ti_3C_2$ antenna is tested by transmitting movie trailers through the wireless communication platform. When the antenna is flat, the point of the planisphere is very dense, meaning the high communication quality, and the bit error rate is almost zero (The left picture in Fig. 4l and Supplementary Movie 2). Currently, the transmitted movie is very clear. When the $Ti_3C_2$ antenna is bent horizontally or vertically, the points on the planisphere are slightly scattered, and the bit error rate at this time is still closed to zero, and the movie still transmits normally (The middle and right picture in Fig. 4l and Supplementary Movie 3). The detailed bending radii effects further indicate that the real-time movie transmission and reception of the $Ti_3C_2$ antennas can still be realized under the maximum bending angle (>200º), demonstrating the advantages of mechanical flexibility (Supplementary Fig. 25 and Supplementary Movie 4). The transmission effect can also be achieved when the antennas are in the non-line of sight (Supplementary Fig. 26 and Supplementary Movie 5) or at different orientation angles (Supplementary Fig. 27 and Supplementary Movie 6). The real-time communication can also proceed in the long-range distance of 1-5 m (Supplementary Fig. 28 and Supplementary Movie 7). The communication between the antennas where one antenna transmits a signal generated by the signal generator and the receiving antenna reveals the response has been revealed in a spectrum analyzer (Supplementary Fig. 29). The specific real-time application of the flexible ultrawideband $Ti_3C_2$ monopole antennas is promising in various scenarios including human-computer interaction fields (i.e., smart medical treatment, individual combat, etc), IoT (i.e., real-time sensing, identity recognition, near-field communication, etc), mobile communication systems, large data transfer, video calls, multi-person online conferences, and information exchange of large data volumes. The humidity effect and heat effect on the antenna performance have been evaluated through the implementation of $Ti_3C_2$ antennas as radiating and sensing elements while the antenna sensor is connected to the

vector network analyzer, showing the sensing potentials in cutting-edge IoT applications (Supplementary Figs. 30 and 31).

## Discussion

In summary, this work reports the elliptical ultrawideband $Ti_3C_2$ monopole antenna via high-resolution extrusion printing technology. The bandwidth of 1.7-4.0 GHz in the working frequency band covers WLAN, Bluetooth, and 5 G (n41, n78) frequency bands, which is comparable to the traditional Cu antenna and superior to previously reported $Ti_3C_2$ antennas. The molecular glue modification strategy realizes the compact conformal integration of $Ti_3C_2$ layer and F4B dielectric substrate in $Ti_3C_2$ antenna, which overcomes the non-resilience defects of traditional copper antenna. The $S_{11}$ parameter and gain of $Ti_3C_2$ antennas are well maintained after 1000 bending cycles at the bending radii of 1.5, 2.8, and 4.3 cm. The excellent cyclic bending stability ensures fluent real-time wireless transmission for movie trailers in bending states, which is also the first demo instance of $Ti_3C_2$ antenna in recently reported works. This work presents a significant microelectronic printing technological advance in developing commercial antenna with excellent flexibility and ultrawideband for efficient wireless data communication and transmission at fast-growing IoT applications.

## Methods

### Materials

$Ti_3AlC_2$ powder (99.99 wt%) was purchased from 11 Technology Co., Ltd. LiF (99 wt%), dopamine hydrochloride (98 wt%) and tris-magnesium buffer (1 M, pH = 8.5) were purchased from Aladdin Biochemical Technology Co., Ltd. Hydrochloric acid (HCl) solution was purchased from Nanjing Chemical Reagent Co., Ltd. The dielectric substrate (F4B220M) was purchased from Shenzhen Dongxin Jiuzhou Technology Co., Ltd. All materials in this work were used as received without further purification.

### Characterization

SEM image was characterized by scanning electron microscopy (FESEM, Hitachi S-4800). TEM image was measured by transmission electron microscopy (Hitachi HT7700). XRD pattern was carried out by X-ray diffractometer (Bruker AXS D8 Advance), using Cu $K_\alpha$ radiation ($\lambda = 1.5406$ Å) over the range of $2\theta = 5.0 \sim 60.0°$. FT-IR spectroscopy was obtained using FT-IR spectrophotometer (PerkinElmer Spectrum Two). XPS was performed on Thermo ESCALAB 250XI. Thin-film sheet resistance was tested by dual electric digital four-probe tester (ST2263, China). $S_{11}$ and $S_{21}$ of all TLs were measured by vector network analyzer (Rohde & Schwarz ZVA67). Radiation efficiency was measured in a SATIMO anechoic chamber. Signal transmission performance was carried out by the Universal Software Radio Peripheral (NI USRP-2943R). CST Microwave Studio, a full wave time domain finite integration method solver, was used to model $Ti_3C_2$ and copper microstrip TLs, and ultrawideband monopole antennas. In the actual measurement, Anritsu 3680k microwave test fixture was used instead of a pressed connector.

### Preparation of m-$Ti_3C_2$ bulks

First, LiF (0.5 g) was dissolved in HCl (10 mL, 9 M). Then, the commercial $Ti_3AlC_2$ bulks (0.5 g) were slowly added to the mixture and conserved at 60 °C for 24 h. Afterward, the products were washed with deionized water five times until the pH was above 6. Finally, the samples were dried under a vacuum for 12 h.

### Preparation of $Ti_3C_2$ ink

m-$Ti_3C_2$ bulks (100 mg) were dispersed in 10 mL deionized water and sonicated (60 kHz, 360 W) for 1 h. Then, the dispersion was centrifuged at 3500 rpm for 1 h. 80% of the upper solution was sucked up to discard the unexfoliated m-$Ti_3C_2$. Finally, the solution was subsequently centrifuged at 5000 rpm for 1 h to collect the exfoliated $Ti_3C_2$ nanosheets.

### Treatment of dielectric substrate

$Ti_3C_2$ microstrip TLs were designed with a characteristic impedance of 50 Ω to transmit electromagnetic waves. The commercial F4BM220 dielectric substrate with a dielectric constant of 2.2 was chosen, in which the top copper-clad layer was etched away and the bottom copper-clad layer was retained. The dopamine hydrochloride (400 mg) was dissolved in deionized water (100 mL) and tris-magnesium buffer (30 mL) was diluted in deionized water (70 mL). The above solution was fully mixed in an open system, and the pH value was adjusted to 8.5 at 40 °C. The pretreated dielectric substrate was immersed in the above mixture for 24 h under stirring. After the reaction was completed, the impurity on the surface was constantly washed away.

### Design and simulation of $Ti_3C_2$ and copper TLs

$Ti_3C_2$ microstrip TLs are composed of three layers, including the dielectric layer (F4BM220), the conductor $Ti_3C_2$ layer, and the ground plane (copper). $Ti_3C_2$ TL with 0.8 mm width and 30 mm length was designed for testing. The pressed connector was modelled with a wave port excitation. In order to minimize the reflection coefficient, a copper microstrip TL with a width of 0.8 mm and a thickness of 18 μm was designed to match the system characteristic impedance of 50 Ω. In the testing scenario, the electromagnetic waves ranged from 1 to 10 GHz. Copper microstrip TL was fabricated externally by manufacturers using standard printed circuit boards fabrication process.

### $Ti_3C_2$ microstrip TLs measurement

Anritsu 3680k microwave test fixture was used for the interconnection between $Ti_3C_2$ TL and vector network analyzer cable. Scattering parameters of TL were measured using a vector network analyzer (Rohde & Schwarz ZVA67). The attenuation constant ($\alpha$) of $Ti_3C_2$ microstrip TL with the length of 30 mm was evaluated based on the S-parameters. The Eq. (1) was as follows.

$$\alpha = \frac{1}{l} 10 \lg\left(\frac{1 - |S_{11}|^2}{|S_{21}|^2}\right) \tag{1}$$

where $l$ (mm) is the length of TL, $S_{11}$ is reflection coefficient, and $S_{21}$ is transmission coefficient.

### Design and simulation of ultrawideband monopole antenna

$Ti_3C_2$ antennas and Cu antennas were modeled by considering the skin depth, surface roughness and conductivity of the conductor. The specific size of the elliptical ultrawideband monopole antenna was determined by the following Eqs. (2) and (3)[48].

$$f_L = \frac{7.2}{[(L + r + p) \times k]} \tag{2}$$

$$2 \times \pi \times r \times L = \pi \times a \times b \tag{3}$$

Where k is taken as 0.823 empirically for a dielectric layer with $\varepsilon_r = 2.2$ and $h = 0.254$ mm. $L$ is the long axis of the ellipse, $b = L/2$, and $r$ is the effective radius of an equivalent cylindrical monopole antenna. $p$ is the length of the 50 Ω feed line when the TL width is 0.8 mm. We first assume that $L = 3.9$ cm, $p = 0.1$ cm, and $r \approx 0.375$ cm can be determined by Eq. (2). Then, the value of $a \approx 1.5$ cm can be determined by Eq. (3).

By adjusting the axial ratio of the ellipse and the design of the ground plane, the bandwidth of the antenna can be increased. The antenna was simulated using a time domain solver. The 3D radiation pattern of $Ti_3C_2$ antenna was also drawn using CST as shown in Supplementary Fig. 18. As the frequency increases, 3D pattern of the

antennas had no obvious changes, maintaining an omnidirectional radiation state.

## Antenna measurement
**Radiation efficiency measurement.** It was performed in a SATIMO anechoic chamber.

**Gain measurement.** It was performed in a far-field anechoic chamber. The accurate gain of ultrawideband $Ti_3C_2$ monopole antennas in a far-field anechoic chamber was measured using gain-transfer (gain-comparison) method[49]. The transmitting antenna and the receiving antenna facing each other at the same height were separated by a certain distance. The standard transmitting antenna in this experiment was a log-periodic antenna for 0.5-6 GHz (A-INFO DS-50600, Chengdu, China). The receiving antennas in this experiment included two parts. One part of receiving antennas consisted of standard gain horn antennas of 1.7–2.6 GHz, 2.6–3.95 GHz, and 3.95–5.85 GHz (A-INFO LB-430-10, LB-284-10, and LB-187-15, Chengdu, China). The gain ($G_{REF}$) of a standard gain horn antenna was known from the antenna manual provided by antenna manufacturers. The other part of receiving antennas consisted of a $Ti_3C_2$ antenna (i.e., $Ti_3C_2$–3 μm, $Ti_3C_2$–5.5 μm, and $Ti_3C_2$–10 μm) and a copper antenna. A laser level was used to calibrate the height of the transmitting and receiving antennas. The antenna test software (AT Studio) that matched with the far-field anechoic chamber was used to measure the realized gain.

The transmitting frame and receiving a frame of the antenna were shown in Supplementary Fig. 13, and the receiving frame can be rotated 360° around the Z axis. The transmitting antenna was set up and excited with a power of 10 dBm. The receiving electrical level ($E_{REF}$) of the horizontal polarization direction and the vertical polarization direction of the standard gain horn antenna were tested first. Then the receiving electrical level ($E_{AUT}$) of the horizontal polarization direction and the vertical polarization direction of a $Ti_3C_2$ antenna (i.e., $Ti_3C_2$–3 μm, $Ti_3C_2$–5.5 μm and $Ti_3C_2$–10 μm) and a copper antenna were tested. Finally, the realized gain of the $Ti_3C_2$ antenna was calculated by the following Eq. (4).

$$G_{AUT} = (E_{AUT} - E_{REF}) + G_{REF} \qquad (4)$$

**Radiation pattern measurement.** It was performed in a far-field anechoic chamber. The radiation pattern of $Ti_3C_2$ antenna was measured in the far-field anechoic chamber. The system can scan 360° of the antenna. For scanning, it is necessary to ensure that the transmitting and receiving antennas maintained the same polarization direction. The normalized radiation pattern can be obtained by simply processing the measured radiation pattern. Data was recorded by 0.1° sweeping azimuth ($\varphi$) and 0.1° sweeping roll ($\theta$).

**Humidity effect measurement.** The humidity effect on the antenna performance has been evaluated through the implementation of $Ti_3C_2$ antenna as radiating and sensing element while the antenna sensor is connected to the vector network analyzer (Supplementary Fig. S30a). The antenna sensor is positioned inside the sealed custom-made chamber, and the humidity is pumped by the humidifier. The resonant peak is used as the initial frequency to measure the resonant frequency shift during humidity sensing measurement (Supplementary Fig. S30b,c).

**Heat effect measurement.** The heat effect on the antenna performance has been measured through the implementation of $Ti_3C_2$ antenna as radiating and sensing elements while the antenna sensor is connected to the vector network analyzer (Supplementary Fig. S31a). The antenna is irradiated by an infrared lamp (317 mW cm$^{-2}$) as a heat source, and the thermal imager is used to monitor the real-time temperature (Supplementary Fig. S31b).

## Data availability

The datasets generated during and/or analyzed during the current study are available from the corresponding author on reasonable request. Source data are provided with this paper.

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

## Acknowledgements

This work was supported by National Funds for Distinguished Young Scientists (61825503 (Q.Z.)), National Natural Science Foundation of China (62174086 (W.Z.), 62001250 (L.L.), 62288102 (W.H.), and 61772287 (F.T.)), and Postgraduate Research & Practice Innovation Program of Jiangsu Province (SJCX22_0253 (F.L.)).

## Author contributions

W.Z. and H.N. contributed equally to this work. L.L., W.H., and Q.Z. conceived the idea and supervised the project. W.Z. designed the project and wrote the manuscript. H.N. co-wrote the manuscript and performed the device measurement. C.D. carried out the device fabrication and characterization. F.Q., F.L., F.T., and P.Y. performed ink development and characterization experiments. S.L., W.He., and X.W. performed materials synthesis. All authors contributed to analyze the data.

## Competing interests

The authors declare no competing interests.
