## [Peer Review File · Nature Communications]

REVIEWER COMMENTS

Reviewer #1 (Remarks to the Author):

A 2D Titanium Carbide Printed Flexible Ultrawideband monopole antenna for wireless communication is proposed. It is highly appreciated that much effort has been put and it is evident in the measurements performed. Following comments for the authors to take into consideration before a revision is submitted.

1. There is lack in clarity of dimensions of the designed antenna
2. Why circular patch of given size is chosen. It is suggested to provide the evolution of the design with respect to the reflection coefficient performance and material properties in comparison with the impedance matching may also be discussed.
3. Provide the units in Fig.4.f and also mention the plane (E-plane and H-plane) for the figures.
4. Provide the comparison of the proposed work with the previous literature both quantitative and method of realisation way.
5. Highlight the novelty of your work in key points and it is suggested to discuss regarding the specific real-time application where the antenna can be used.
6. What are the realistic constraints of the proposed flexible antenna with the new material? Also discuss the bending radii effects and upto how much bent radius the antenna would perform without any real time staggering of information transmittance or reception. Suggested to perform further experiments to investigate the bending aspects in detail.

Good work and Overall a revision is recommended to meet the standards of Nature Communications.

Reviewer #2 (Remarks to the Author):

The manuscript on the fabrication and characterization of a flexible ultrawide monopole antenna is very interesting and well written.

Below there are some considerations for improving the paper.

1) It is not clear from reading the introduction of the paper what the innovative aspect of your fabrication approach is and why it appears to be beneficial to existing approaches in the literature.

2) Why was the three-antenna method used for gain estimation? why you don't use two identical antennas fabricated by you? Did you estimate realized gain?

3) Have you made a comparison between simulations and characterization regarding directivity and axial ratio of the antenna?

4) Have you think about of a strategy to increase the antenna gain?

Reviewer #3 (Remarks to the Author):

This work reports an elliptical ultra-wideband Ti3C2 monopole antenna which is fabricated using the extrusion printing method. The author initially used different thicknesses of MXene and measured the resistivity for them Later they used thicknesses of 3, 5.5, 10 μm to fabricate MXene-based antennas. The antenna demonstrated a relative bandwidth of 1.7-4.0 GHz. The results demonstrated the flexibility of the antenna in different bending cases for different bending radii. The authors used the antennas to demonstrate communication in a short range of operations. The authors present the work for wireless data communication in fast-growing IoT applications.

Based on my review I would suggest a major revision for this work.

Comments:

1- MXene film-based antenna were first introduced in "Appl. Mater. Today, vol. 26, p. 101294, Mar. 2022, doi: 10.1016/j.apmt.2021.101294." and they were used for communicating for a second antenna in the range of meters. Second, the MXene-based antennas were also introduced for

wireless gas sensing application in “Adv. Mater. Interfaces, p. 2102411, Mar. 2022, doi: 10.1002/ADMI.202102411.”. Based on this I don’t find the purposed work novel as the design and implementation of MXene-based antenna’s have been performed in the previous studies.

2- In Fig.4 why does using a lower thickness of MXene membrane cause a deeper notch between 2 to 2.5 GHz?

3- Are the designed MXene antennas optimized? I would suggest that the authors present their simulation studies for the design.

4- The authors presented antennas with a gain below 4 dB and radiation efficiency below 80% What can be done to increase the gain and the radiation efficiency of the antenna?

5- In “Adv. Mater., vol. 33, no. 1, pp. 1–7, 2021, doi: 10.1002/adma.202003225.” the authors purposed antennas with higher radiation efficiency a. What would be the advantage of your work to theirs?

6- Does humidity or heat affect the antenna’s performance over time?

7- Can the purposed antenna be used for sensing applications which are a top trend in IoT studies? If yes can the antenna itself act as a sensor? If yes, what can be the sensing application for the purposed antenna?

8- Figure 4l shows the transmission of data between two MXene-based antennas. Can this communication also happen in the long-range in the range of meters? Can the authors present the results of this communication? Also, I am interested to see the communication between the antenna where one antenna transmits a signal generated by a signal generator and the receiving antenna reveals the response in a spectrum analyzer

9- Are the communication measurement results repeatable over time?

10- Can the antennas communicate when they are not in the line of sight or have different orientation angles?

11- The authors are encouraged to present a table for the thickness of MXene, the conductivity and the skin depth of the MXene which they used.

12- The authors mentioned for different bending angles that “The bandwidth and center frequency can be well maintained and the gain differences fluctuate within ± 1.0 dBi.”. This gain error margin is high for an antenna with a gain of 4 dB.

Point-by-Point Response to Referees

Reviewer #1 (Remarks to the Author): A 2D Titanium Carbide Printed Flexible Ultrawideband monopole antenna for wireless communication is proposed. It is highly appreciated that much effort has been put and it is evident in the measurements performed. Following comments for the authors to take into consideration before a revision is submitted.

Response: We would first like to thank the reviewer for the positive comment on the significance and quality of our work. Your insightful comments are very constructive for the further improvement of our work. We have tried our best to revise our manuscript accordingly.

1. There is lack in clarity of dimensions of the designed antenna.

Answer: We are very grateful for the reviewer’s wise suggestion. The dimensions of the designed antenna have been demonstrated in the revised Supplementary Fig. 11.

Supplementary Fig. 11 | The dimension and geometry of ultrawideband elliptical Ti_3C_2 monopole antennas working in the frequency of 1.7-4.0 GHz.

2. Why circular patch of given size is chosen. It is suggested to provide the evolution of the design with respect to the reflection coefficient performance and material properties in comparison with the impedance matching may also be discussed.

Answer: We would like to thank the reviewer for this insightful suggestion. The given size of the circular patch is shown in Supplementary Fig. 11 in the revised Supporting Information. The design principles are as follows.

In 1992, Honda first proposed an ultrawideband circular monopole antenna with an extraordinary impedance bandwidth and omnidirectional radiation performance due to the symmetrical current distribution on the disk (*Proc. Int. Symp. Antennas Propag.*, 4, 1145 (1992)). In comparison with the circular patch, the elliptical patch has similar symmetrical currents and improved impedance bandwidth due to the better impedance matching performance (*IEEE Trans. Antennas Propag.* 46, 294-295 (1998)) (See the notes in Supplementary Fig. 12 in the revised Supporting Information).

Supplementary Fig. 12 | The design principles of the ultrawideband elliptical Ti_3C_2 monopole antennas. a, Simulated S_{11} of the Ti_3C_2 antennas with different axial ratios and measured S_{11} of the Ti_3C_2 antennas with axial ratio of 1.3. **b**, Comparison of the measured and simulated S_{11} of the Ti_3C_2 antennas by the parameters of conductivity and sheet resistance.

For the design of an ultrawideband monopole antenna, the specific size of the antenna is determined by the following equations (2) and (3).

$$f_L = \frac{7.2}{[(L+r+p) \times k]} \text{ GHz} \quad (2)$$

$$2 \times \pi \times r \times L = \pi \times a \times b \quad (3)$$

Where k is taken as 0.823 empirically for a dielectric layer with $\epsilon_r = 2.2$ and $h = 0.254$ mm. L is the long axis of the ellipse, $b = L/2$, and r is the effective radius of an equivalent cylindrical monopole antenna. p is the length of the 50 Ω feed line. We first assume that $L = 3.9$ cm, $p = 0.1$ cm, and $r \approx 0.375$ cm can be determined by equation (2). Then, the value of $a \approx 1.5$ cm can be determined by equation (3) (See the experiment section “**Design and simulation of ultrawideband monopole antenna**” in the revised manuscript).

In order to obtain the widest impedance bandwidth, the axial ratio of the elliptical Ti₃C₂ antennas is first analyzed and the simulated reflection coefficient S_{11} is shown in Supplementary Fig. 12a in the revised Supporting Information. When the axial ratio of the ellipse is less than 1.3, the relative bandwidth of the antennas widens with the increase of axial ratio. S_{11} of the antennas is greater than -10 dB in 3-4 GHz. However, when the axial ratio of the ellipse is greater than 1.3, the relative bandwidth of the antennas is almost unchanged. Considering the miniaturization requirement of the antennas, the axial ratio of the ellipse is set as 1.3. (See the notes in Supplementary Fig. 12 in the revised Supporting Information)

The material system and full-wave simulation are combined based on sheet resistance for the first time to accurately predict the performance of Ti₃C₂-10 μm antennas (Supplementary Fig. 12b). In comparison with conductivity, the sheet resistance is more accurate for full-wave simulation in Ti₃C₂ microwave performance analysis. The actual bandwidth of Ti₃C₂-10 μm antennas is consistent with the simulated bandwidth. Thus, the design rationality of antenna size is verified (See the notes in Supplementary Fig. 12 in the revised Supporting Information).

In comparison with the poor flexibility and sophisticated manufacturing process of metal, the flexibility, scalability, and ease of solution processing of Ti₃C₂ nanosheets

are promising for lightweight and flexible antenna components to support the rapid development of integrated RF devices. Ti_3C_2 antennas have the potentials to be an alternative to conventional metal antennas for efficient and reliable power delivery at higher frequencies in RF and 5G communication.

3. Provide the units in Fig. 4.f and also mention the plane (E-plane and H-plane) for the figures.

Answer: We appreciate the reviewer’s wise comment. The unit is dBi. The left pattern in Fig. 4f is E-plane and the right pattern in Fig. 4f is H-plane as shown in Fig. 4f in the revised manuscript.

Fig. 4f. Typical radiation pattern of Ti_3C_2 -5.5 μm antennas measured in the anechoic chamber. The unit is dBi.

4. Provide the comparison of the proposed work with the previous literature both quantitative and method of realisation way.

Answer: We appreciate the reviewer’s wise comment. Both quantitative performance and preparation method of patch antennas have been comprehensively compared, as shown in Supplementary Table 3 in the revised Supporting Information.

Supplementary Table 3. The quantitative performance and preparation method of patch antennas made of different materials.

Materials	Thickness [μm]	Efficiency [%]	Frequeny [f_0 ; GHz]	Substrate	Substrate thickness [mm]	Conductivity [S cm^{-1}]	Gain [dB]	Method	Ref.
Graphene	10	60	4.8	Kapton	0.076	/	2.3	Screen printing	16
Graphene	25	64.9	6	PDMS	2	/	/	Low temperature fabrication process	17
Cu mesh	20	49-56.88	2.4-2.5	Acrylic plate	1.2	10000	2.65	Vacuum evaporation	18
Copper	35	50.93	2.5				4.77	Vacuum evaporation	
Cu mesh	5	42.69	2.45	Acrylic	1	/	2.63	Physical vapor deposition	19
IZTO/Ag/IZTO	0.1	7.76	2.45			/	-4.23	Physical vapor deposition	
Ag/Ni/Cu fabric	130	58.6	2.45	PDMS	3	/	4.16	Multilayer printing	20
Silver nanoparticle	3	11	2.45	Cardboard	0.56	200000	1	Inkjet-printing	21
Silver paste	26.52	2.81	2.45	NinjaFlex	1.2	170	-7.2	3D printing	22
Silver nanowire	500	41.83	2.92	PDMS	1	8130	4.9	Screen printing	23
Silver	3	31.6	1.89	Kapton	0.1	/	2.2	Inkjet-printing	24
EGaIn	100	45-60	3.43	PDMS	1	34000	/	Injecting method	25
EGaIn	1500	75	5.2	Photopolymer resin	6	51000	/	3D printing	26
Ti ₃ C ₂	1.0	80-90.4	5.6	RT 5880	1.6	15000	5.74	Spray coating	14
	3.2	87-98.4	10.9				7.14		
	5.3	90.6-99	16.4				5.48		
Ti ₃ C ₂	20	/	1.841	Rogers 6010	1.91	4200	1.9	Vacuum-assisted	27

								filtration	
Ti ₃ C ₂	3	/	1.834	Rogers 6010	1.91	/	2	Vacuum-assisted filtration	7
	3	68.4				7765.84	3.14		
Ti ₃ C ₂	5.5	68.7	2.45	F4B	0.254	7221.39	3.04	Extrusion printing technology	This work
	10	76.5				4474.07	3.23		k

*EGaIn: Eutectic Gallium Indium (liquid metal); PET: Polyethylene Terephthalate; IZTO: In–Zn–Sn–O.

5. *Highlight the novelty of your work in key points and it is suggested to discuss regarding the specific real-time application where the antenna can be used.*

Answer: We sincerely appreciate the reviewer for the thoughtful recommendation.

The novelty of this work is highlighted in the following key points in the revised manuscript. (i) This work first exploits the flexible ultrawideband Ti₃C₂ monopole antenna by combining strategies of interfacial modification and advanced extrusion printing technology (Page 2, line 6~7). (ii) In order to achieve the compact integration, the polydopamine, as “molecular glue” nano-binder, acts as a secondary platform to improve the interfacial adhesion interactions between dielectric substrate and printed Ti₃C₂ film, contributing the conformal integrated Ti₃C₂ antennas with the high spatial uniformity and mechanical flexibility (Page 6, line 7~10). For comparison, the copper monopole antenna with the same structure has no resilience at any bending angle (Page 12, line 5~6). (iii) The bandwidth and center frequency of Ti₃C₂ antenna can be well maintained and the gain differences fluctuate within ±0.2 dBi at the low frequency range after the bent antenna returns to the flat state (Page 2, line 10~12). The excellent cyclic bending stability ensures the fluent real-time wireless transmission for movie trailers in bending states, which is also the first demo instance of Ti₃C₂ antenna in recently reported works (Page 15, line 1~3). The transmission effect can also be achieved when the antennas are in the non-line of sight or at different orientation angles. (Page 13, line 17~20) (iv) The bandwidth of 1.7-4.0 GHz in working frequency band

covers WLAN, Bluetooth, and 5G (n41, n78) frequency bands, which is comparable to the traditional Cu antenna and superior to previously reported Ti₃C₂ antennas (Page 14, line 15~18). (v) The macroscopic electromagnetic property has been analyzed and applied to the simulation design of ultrawideband Ti₃C₂ monopole antennas. The material system and full-wave simulation are combined based on sheet resistance for the first time to accurately predict the antenna performance (Page S15, line 3~4).

According to Shannon's equation,

$$C = B \times \log_2 \left(1 + \frac{S}{N} \right) (\text{bit} / \text{s})$$

when the signal-to-noise ratio (S/N) is the same, a larger bandwidth (B) can provide a larger data transmission rate (C). The specific real-time application of the flexible ultrawideband Ti₃C₂ monopole antennas in this work is promising in various scenarios including human-computer interaction fields (i.e., smart medical treatment, individual combat, etc), IoT (i.e., real-time sensing, identity recognition, near-field communication, etc), mobile communication systems, large data transfer, video calls, multi-person online conferences, and information exchange of large data volumes. (Page 14, line 2~7)

6. What are the realistic constraints of the proposed flexible antenna with the new material? Also discuss the bending radii effects and upto how much bent radius the antenna would perform without any real time staggering of information transmittance or reception. Suggested to perform further experiments to investigate the bending aspects in detail.

Answer: We sincerely appreciate the reviewer for the insightful suggestion.

As a novel family of two-dimensional (2D) materials, Ti₃C₂ has shown excellent properties for the application of proposed flexible antennas due to their comprehensive performance of electrical conductivity, mechanical stability, high flexibility, and easy processability. However, the present research on the wireless communication performance of Ti₃C₂ antennas is still in the exploratory stage, and there are still the realistic constraints for the proposed flexible antennas before advancing to full-fledged

manufacturing and practical applications. (i) The oxidation of Ti_3C_2 will lead to the decreased conductivity, especially in the case of few- or mono- layer nanosheets. The encapsulation will be an effective strategy to inhibit the oxidation and improve the long-term chemical stability of Ti_3C_2 antenna (*Nat. Commun.*, **13**, 3223 (2022); *Nano-Micro Lett.* **13**, 115 (2021)). (ii) The flexible transparent antenna is still desirable in specific practical application scenarios. The trade-off between optical transparency and high transmission performance can be achieved through increasing the electrical conductivity (*Adv. Mater. Technol.* 2101277 (2022)). (iii) The transmission mechanism of electromagnetic waves in Ti_3C_2 antennas is not very clear. For example, when the thickness of Ti_3C_2 film is less than the skin depth, the excellent antenna transmission performance can be achieved. The relevant principle needs to be further studied (*Sci. Adv.* **4**, eaau0920 (2018)).

Supplementary Fig. 25 | Real-time video transmission of Ti_3C_2 antennas under the flat states or different bending angles of (a) 0° , (b), 90° , (c), 180° , and (d), $> 200^\circ$.

The bending radii effects of Ti_3C_2 antennas have been systematically tested under different bending angles for the real-time information transmission and reception (Supplementary Fig. 25 and Supplementary Video 4). When the transmitting antenna and receiving antenna are both in flat state, the video transmission is very stable and fluent (Supplementary Fig. 25a). When the receiving antenna is bent at 90° , the video transmission can be well maintained (Supplementary Fig. 25b). As the receiving antenna is bent at 180° , the video transmission is slightly weakened (Supplementary Fig. 25c). When the maximum bending angle ($> 200^\circ$) is reached, the video signal becomes seriously delayed, but the signal transmission can still be realized (Supplementary Fig. 25d). The reason for this phenomenon is that the radiation pattern and current distribution of the antennas is distorted as the bending angle increases, which leads to the attenuated video transmission. Significantly, the transmission performance can return to the initial fluent state after the antenna returns from the bent state to the flat state. (See the notes in Supplementary Fig. 25 in the revised Supporting Information)

Good work and Overall a revision is recommended to meet the standards of Nature Communications.

Answer: We sincerely appreciate the reviewer for the positive comment on the significance and quality of our work. We have tried our best to revise our manuscript accordingly and addressed all the comments point by point.

Reviewer #2 (Remarks to the Author): The manuscript on the fabrication and characterization of a flexible ultrawide monopole antenna is very interesting and well written. Below there are some considerations for improving the paper.

Answer: We are sincerely grateful for your positive comments on our work. Your suggestions are professional and constructive, and we have tried our best to revise our manuscript accordingly.

1) It is not clear from reading the introduction of the paper what the innovative aspect of your fabrication approach is and why it appears to be beneficial to existing approaches in the literature.

Answer: We are very grateful for this valuable comment. The innovative aspect of the extrusion printing approach has been stated in the introduction in the revised manuscript (Page 4, line 6~15). As one of the representative direct ink printing protocols, the extrusion printing technique has been a revolutionary and eco-friendly manufacturing route for mass production of flexible integrated electronics with the high-resolution geometry pattern and digital customization (*Adv. Mater.* **32**, 1908486 (2020); *Adv. Energy Mater.* **9**, 1901839 (2019); *Nat. Commun.* **11**, 5543 (2020)). It not only can generally deposit the functional viscoelastic inks with a large concentration window and suitable fluidic properties (e.g., surface tension and viscosity) under ambient conditions (*Nanoscale* **12**, 19007-19042 (2020); *Sci. Adv.* **6**, eaba5029 (2020); *Adv. Mater. Interfaces* **8**, 2101175 (2021)), but also has apparent advantages in realizing high-precision conformal printing on different substrates (whether flat or curved) without additional masks and accessories, as well as avoiding time-consuming and complicated transfer process, which is superior to previously reported screen printing, physical vapor deposition, and spray coating, etc (*Nat. Commun.* **13**, 3223 (2022); *IEEE Antennas Wirel. Propag. Lett.* **16**, 1883-1886 (2017); *IEEE Antennas Wirel. Propag. Lett.* **16**, 772-775 (2017); *Adv. Mater.* **33**, 2003225 (2021)).

This work first exploits the direct extrusion printing technology of additive-free concentrated Ti_3C_2 inks for flexible ultrawideband Ti_3C_2 monopole antenna. The polydopamine (PDA) is chosen as “molecular glue” nano-binder between Ti_3C_2 film

and dielectric substrate, contributing the conformal integrated microstrip TLs and antennas with the high spatial uniformity and mechanical flexibility. (Page 5, line 5~9)

2) *Why was the three-antenna method used for gain estimation? why you don't use two identical antennas fabricated by you? Did you estimate realized gain?*

Answer: We are very grateful for the reviewer's wise suggestion. In order to facilitate the reader's understanding, the "three-antenna method" has been changed to "gain-transfer (gain-comparison) method" in the revised manuscript (Page 19, line 2). Gain-transfer (gain-comparison) method has been written into the IEEE standard in 1952, and has been commonly used to measure the gain of an antenna (Balanis, C. A. *Antenna theory: Analysis and design* (John Wiley & sons, 2015)). During the testing process, we don't use two identical antennas fabricated by us. But we use two standard antennas and one fabricated Ti₃C₂ antenna. The standard antennas are all purchased from antenna manufacturers, and have been strictly calibrated with a standard gain manual before leaving the factory. The anechoic chamber is equipped with a laser level for alignment during the measurement process to get the most accurate gain data. The realized gain has been estimated and verified by both full-wave simulation and measurement in the anechoic chamber.

The detailed gain measurement is as follows (See the experiment section "**Gain measurement**" in the revised manuscript). The transmitting antenna and the receiving antenna facing each other at the same height were separated by a certain distance. The standard transmitting antenna in this experiment was a log-periodic antenna for 0.5-6 GHz (A-INFO DS-50600, Chengdu, China). The receiving antennas in this experiment included two parts. One part of receiving antennas consisted of standard gain horn antennas of 1.7-2.6 GHz, 2.6-3.95 GHz and 3.95-5.85 GHz (A-INFO LB-430-10, LB-284-10 and LB-187-15, Chengdu, China). The gain (G_{REF}) of a standard gain horn antenna was known from the antenna manual provided by antenna manufacturers. The other part of receiving antennas consisted of a Ti₃C₂ antenna (i.e., Ti₃C₂-3 μ m, Ti₃C₂-5.5 μ m and Ti₃C₂-10 μ m) and a copper antenna. A laser level was used to calibrate the

height of the transmitting and receiving antennas. The antenna test software (AT Studio) that matched with the far-field anechoic chamber was used to measure the realized gain.

The transmitting frame and receiving frame of the antenna were shown in the Supplementary Fig. 13, and the receiving frame can be rotated 360° around the Z axis. The transmitting antenna was set up and excited with a power of 10 dBm. The receiving electrical level (E_{REF}) of the horizontal polarization direction and the vertical polarization direction of the standard gain horn antenna were first tested. Then, the receiving electrical level (E_{AUT}) of the horizontal polarization direction and the vertical polarization direction of a Ti_3C_2 antenna (i.e., Ti_3C_2 -3 μm , Ti_3C_2 -5.5 μm and Ti_3C_2 -10 μm) and a copper antenna were tested. Finally, the realized gain of the Ti_3C_2 antenna was calculated by the following equation (4).

$$G_{AUT} = (E_{AUT} - E_{REF}) + G_{REF} \quad (4)$$

3) *Have you made a comparison between simulations and characterization regarding directivity and axial ratio of the antenna?*

Answer: We are very grateful for the reviewer's wise suggestion.

The comparison between simulations and characterization regarding directivity of the Ti_3C_2 antenna has been made. The directivity of an antenna is equal to the ratio of the maximum power density to its average value over a sphere as observed in the far field of an antenna (Kraus, J. D. & Marhefka, R. J. *Antennas: For All Applications, Third Edition*). The gain of an antenna is an actual or realized quantity which is less than the directivity due to ohmic losses in the antenna. The relationship between directivity and gain can be derived according to the equation (S1),

$$D = G / k \quad (S1)$$

where D is the directivity of the antenna, G is the gain of the antenna, k is the radiation efficiency of the antenna. The measured and simulated directivity of Ti_3C_2 antennas is shown in Supplementary Fig. 16 in the revised Supporting Information. The difference in directivity between Ti_3C_2 antennas and copper antenna is very small. The results show that using Ti_3C_2 instead of copper does not cause the decreased radiation

angle of the antenna. The measured directivity is slightly larger than the simulated directivity, which is due to the influence of metal connector. (See the notes in Supplementary Fig. 16 in the revised Supporting Information)

Supplementary Fig. 16 | Measured and simulated directivity of Ti_3C_2 antennas.

In addition, the comparison between simulations and characterization of radiation patterns has been shown in Supplementary Fig. 17 in the revised Supporting Information.

Supplementary Fig. 17 | Comparison of measured and simulated normalized radiation patterns (E/H plane) of antenna under different operating frequencies. a, 2.0 GHz, b, 2.4 GHz, c, 2.8 GHz, d, 3.2 GHz, e, 3.6 GHz, f, 4.0 GHz, respectively.

The comparison between simulations and characterization of axial ratio has also been made. The axial ratio of the antenna is defined as the ratio of the length of the major axis and minor axis of the ellipse. In order to obtain the widest impedance bandwidth, the axial ratio of the elliptical Ti_3C_2 antennas is first analyzed and the simulated reflection coefficient S_{11} is shown in Supplementary Fig. 12a in the revised Supporting Information. When the axial ratio of the ellipse is less than 1.3, the relative bandwidth of the antennas widens with the increase of axial ratio. S_{11} of the antennas is greater than -10 dB in 3-4 GHz. However, when the axial ratio of the ellipse is greater than 1.3, the relative bandwidth of the antennas is almost unchanged. Considering the miniaturization requirement of the antennas, the axial ratio of the ellipse is set as 1.3. (See the notes in Supplementary Fig. 12 in the revised Supporting Information)

The material system and full-wave simulation are combined based on sheet resistance for the first time to accurately predict the performance of Ti_3C_2 -10 μm antennas (Supplementary Fig. 12b in the revised Supporting Information). In comparison with conductivity, the sheet resistance is more accurate for full-wave simulation in Ti_3C_2 microwave performance analysis. The actual bandwidth of Ti_3C_2 -10 μm antennas is consistent with the simulated bandwidth. Thus, the design rationality of antenna size is verified. (See the notes in Supplementary Fig. 12 in the revised Supporting Information)

Supplementary Fig. 12 | The design principles of the ultrawideband elliptical Ti_3C_2 monopole antennas. a, Simulated S_{11} of the Ti_3C_2 antennas with different axial ratios and measured S_{11} of the Ti_3C_2 antennas with axial ratio of 1.3. **b**, Comparison of the measured and simulated S_{11} of the Ti_3C_2 antennas by the parameters of conductivity and sheet resistance.

From another perspective, the axial ratio related to the radiation performance as the description of electric field vector rotation by time is also considered (Kraus, J. D. & Marhefka, R. J. *Antennas: For All Applications, Third Edition*). However, axial ratio is a significant parameter only in circularly polarized antennas. When the axial ratio is lower than 3 dB in the main lobe direction, the circularly polarized antenna can be mainly used in satellite communications. In general mobile communications, the linearly polarized antennas are commonly used in the multipath environments. Therefore, the antenna proposed in this work is a linearly polarized antenna.

4) *Have you think about of a strategy to increase the antenna gain?*

Answer: We sincerely appreciate the reviewer for this valuable comment. Considering the influence of conductor loss, dielectric loss and equivalent electrical length of the antenna, the maximum gain of 3.2 dBi for Ti_3C_2 - $10 \mu m$ antenna is already within an acceptable range. For a certain of omnidirectionality, the gain of the antenna can be increased through the following two strategies. (i) In terms of materials, the

conductivity of Ti_3C_2 component can be further improved through increasing the large single-layer ratio, narrowing flake size distribution, and reducing layer spacing between the Ti_3C_2 nanosheets (*Nat. Commun.*, **2022**, **13**, 3223). (ii) In terms of antenna design and measurement, we can use alternative substrate with lower dielectric loss (i.e., Rogers 5880), increase antenna orientation and combine multiple antennas into an antenna array (Kraus, J. D. & Marhefka, R. J. *Antennas: For All Applications, Third Edition*).

The according changes have been made in our revised manuscript: “It can be further increased through improving the conductivity of Ti_3C_2 component, using alternative substrate with lower dielectric loss, increasing antenna orientation, or combining multiple antennas into an antenna array.” (Page 11, line 11~14)

Finally, we would like to thank the reviewer again for these valuable comments and for the thoughtful and careful review towards improving our manuscript.

Reviewer #3 (Remarks to the Author): This work reports an elliptical ultra-wideband Ti₃C₂ monopole antenna which is fabricated using the extrusion printing method. The author initially used different thicknesses of MXene and measured the resistivity for them. Later they used thicknesses of 3, 5.5, 10 μm to fabricate MXene-based antennas. The antenna demonstrated a relative bandwidth of 1.7-4.0 GHz. The results demonstrated the flexibility of the antenna in different bending cases for different bending radii. The authors used the antennas to demonstrate communication in a short range of operations. The authors present the work for wireless data communication in fast-growing IoT applications. Based on my review I would suggest a major revision for this work.

Answer: We are sincerely grateful for your positive comments on our work. Your suggestions are professional and constructive, and we have tried our best to revise our manuscript accordingly.

1. MXene film-based antenna were first introduced in “Appl. Mater. Today, vol. 26, p. 101294, Mar. 2022, doi: 10.1016/j.apmt.2021.101294.” and they were used for communicating for a second antenna in the range of meters. Second, the MXene-based antennas were also introduced for wireless gas sensing application in “Adv. Mater. Interfaces, p. 2102411, Mar. 2022, doi: 10.1002/ADMI.202102411.”. Based on this I don’t find the purposed work novel as the design and implementation of MXene-based antenna’s have been performed in the previous studies.

Answer: We are very grateful for the reviewer’s professional comment and wise suggestion.

The first recommended work (*Appl. Mater. Today* **26**, 101294 (2022)) investigates the integration of lightweight Ti₃C₂ circular membranes in microwave resonator and antenna structures. When connected to an oscillator and positioned 2.3 m away from a receiver antenna, the receiver antenna could receive signals from the transmitting Ti₃C₂ patch antenna, which demonstrates the transmission capability of the Ti₃C₂ antenna in a long-range transmission range. The second proposed work (*Adv. Mater. Interfaces* **9**, 2102411 (2022)) investigates the implementation of Ti₃C₂ as radiating and sensing

element to develop a VOC and humidity sensing antenna. The study successfully demonstrates the potentials of the proposed Ti_3C_2 antenna sensor in detecting acetone concentrations (8 ~ 80 ppt) and humidity concentrations ($77.8 \pm 2\% \sim 94 \pm 1\%$). We study these papers carefully and agree with the innovations drawn by the authors. We have cited these papers as important references (i.e., Ref. 10 and Ref. 11 in the revised manuscript) for the widespread application of Ti_3C_2 antenna.

However, several vital challenges still need to be addressed on the construction of flexible Ti_3C_2 antenna for the practical applications. First, the utilization of polyethylene terephthalate and double-sided tape between Ti_3C_2 layer and commercial circuit boards causes the complicated manufacture process as well as hinders the direct conformal integration with flexible electronics and chips, thus leading to the unsatisfactory power delivery and sensitive resonant frequency in wireless communication. Second, the working bandwidth is relatively narrow, and it is difficult to meet the ultrawideband requirements. Third, the lack of interfacial adhesion between the commercial dielectric substrate and additive-free Ti_3C_2 layer impedes the manufacture of flexible Ti_3C_2 antennas capable of compact integration.

To overcome the daunting challenges, our work first exploits flexible ultrawideband Ti_3C_2 monopole antenna prepared through the progressive direct extrusion printing technology. In this work, the novel design and implementation of Ti_3C_2 antennas are shown as follows. (i) This work first exploits the flexible ultrawideband Ti_3C_2 monopole antenna by combining strategies of interfacial modification and advanced extrusion printing technology (Page 2, line 6~7). (ii) In order to achieve the compact integration, the polydopamine, as “molecular glue” nano-binder, acts as a secondary platform to improve the interfacial adhesion interactions between dielectric substrate and printed Ti_3C_2 film, contributing the conformal integrated Ti_3C_2 antennas with the high spatial uniformity and mechanical flexibility (Page 6, line 7~10). For comparison, the copper monopole antenna with the same structure has no resilience at any bending angles (Page 12, line 5~6). (iii) The bandwidth and center frequency of Ti_3C_2 antenna can be well maintained and the gain differences fluctuate within ± 0.2 dBi at the low frequency range after the bent antenna

returns to the flat state (Page 2, line 10~12). The excellent cyclic bending stability ensures the fluent real-time wireless transmission for movie trailers in bending states, which is also the first demo instance of Ti₃C₂ antenna in recently reported works (Page 15, line 1~3). The transmission effect can also be achieved when the antennas are in the non-line of sight or at different orientation angles (Page 13, line 17~20). (iv) The bandwidth of 1.7-4.0 GHz in working frequency band covers WLAN, Bluetooth, and 5G (n41, n78) frequency bands, which is comparable to the traditional Cu antenna and superior to previously reported Ti₃C₂ antennas (Page 14, line 15~18).

According to Shannon's equation,

$$C = B \times \log_2 \left(1 + \frac{S}{N} \right) (\text{bit} / \text{s})$$

when the signal-to-noise ratio (S/N) is the same, a larger bandwidth (B) can provide a larger data transmission rate (C). The specific real-time application of the flexible ultrawideband Ti₃C₂ monopole antennas in this work is promising in various scenarios including human-computer interaction fields (i.e., smart medical treatment, individual combat, etc), IoT (i.e., real-time sensing, identity recognition, near-field communication, etc), mobile communication systems, large data transfer, video calls, multi-person online conferences, and information exchange of large data volumes. (Page 14, line 2~7) (v) The macroscopic electromagnetic property has been analyzed and applied to the simulation design of ultrawideband Ti₃C₂ monopole antennas. The material system and full-wave simulation are combined based on sheet resistance for the first time to accurately predict the antenna performance (Page S15, line 3~4). The experimental reflection coefficient proves that this assumption can accurately predict the actual effect of antennas (Supplementary Fig. 12b). The relative bandwidth of Ti₃C₂ antenna maintains almost the same performance as that of the copper antenna.

Supplementary Fig. 12 | The design principles of the ultrawideband elliptical Ti_3C_2 monopole antennas. b, Comparison of the measured and simulated S_{11} of the Ti_3C_2 antennas by the parameters of conductivity and sheet resistance.

2. In Fig.4 why does using a lower thickness of MXene membrane cause a deeper notch between 2 to 2.5 GHz?

Answer: We are very grateful for the reviewer's professional comment. The bandwidth of the ultrawideband monopole antenna results from the superposition of multiple resonance points, which will cause S_{11} curve to rise upward. As the Ti_3C_2 layer becomes thinner, the local resonance characteristics become stronger. As shown in Fig. 4d in the revised manuscript and Fig. R1, for thinner Ti_3C_2 layer, the sheet resistance of Ti_3C_2 layer becomes larger and the radiation efficiency of the antenna decreases. According to equation (R1) (*IEEE Trans. Microw. Theory Tech.* **23**, 522-526 (1975); *IEEE Trans. Antennas Propag.* **66**, 5180-5192 (2018)), the decreased radiation efficiency of the antenna will lead to the decrease of the average power loss due to the radiation (W_r) in the frequency band of 2.0-2.5 GHz, which will lead to the increased radiation loss (Q_r) value. According to equation (R2) (*IEEE Trans. Microw. Theory Tech.* **23**, 522-526 (1975); *IEEE Trans. Antennas Propag.* **66**, 5180-5192 (2018)), when Q_r increases, Q_t increases and the notch of S_{11} becomes deeper. Thus, as the thickness of Ti_3C_2

components decreases, the correspondingly increased sheet resistance leads to the increased Q_t value and the deeper notch of S_{11} curves. As shown in Fig. R1, the simulation results are consistent with the measurement results.

$$Q_r = \frac{2\pi f_0 U}{W_r} \quad (R1)$$

$$\frac{1}{Q_t} = \frac{1}{Q_r} + \frac{1}{Q_d} + \frac{1}{Q_c} + \frac{1}{Q_s} \quad (R2)$$

where Q_t is the total Q value, Q_r is the radiation loss, Q_d is the dielectric loss, Q_c is the conductor loss, and Q_s is the surface-wave loss.

Fig. R1. Measured and simulated S_{11} parameter of Ti_3C_2 antennas and Cu antennas.

The according changes have been made in our revised manuscript: “As the thickness of Ti_3C_2 components decreases, the deeper notch of S_{11} curves appears due to the stronger local resonance characteristics and decreased radiation efficiency of the antenna.” (Page 10, line 18~21)

3. Are the designed MXene antennas optimized? I would suggest that the authors present their simulation studies for the design.

Answer: We sincerely appreciate the reviewer for the helpful suggestion. The designed Ti_3C_2 antennas are optimized. The simulation studies for the design are presented in the revised Supporting Information.

For the design of an ultrawideband monopole antenna, the specific size of the antenna is determined by the following equations (2) and (3).

$$f_L = \frac{7.2}{[(L+r+p) \times k]} \text{GHz} \quad (2)$$

$$2 \times \pi \times r \times L = \pi \times a \times b \quad (3)$$

Where k is taken as 0.823 empirically for a dielectric layer with $\epsilon_r = 2.2$ and $h = 0.254$ mm. L is the long axis of the ellipse, $b = L/2$, and r is the effective radius of an equivalent cylindrical monopole antenna. p is the length of the 50Ω feed line. We first assume that $L = 3.9$ cm, $p = 0.1$ cm, and $r \approx 0.375$ cm can be determined by equation (2). Then, the value of $a \approx 1.5$ cm can be determined by equation (3) (See the experiment section “Design and simulation of ultrawideband monopole antenna” in revised manuscript).

Supplementary Fig. 12 | The design principles of the ultrawideband elliptical Ti_3C_2 monopole antennas. a, Simulated S_{11} of the Ti_3C_2 antennas with different axial ratios and measured S_{11} of the Ti_3C_2 antennas with axial ratio of 1.3. **b**, Comparison of the measured and simulated S_{11} of the Ti_3C_2 antennas by the parameters of conductivity and sheet resistance.

In order to obtain the widest impedance bandwidth, the axial ratio of the elliptical Ti_3C_2 antennas is first analyzed and the simulated reflection coefficient S_{11} is shown in Supplementary Fig. 12a in the revised Supporting Information. When the axial ratio of the ellipse is less than 1.3, the relative bandwidth of the antennas widens with the increase of axial ratio. S_{11} of the antennas is greater than -10 dB in 3-4 GHz. However, when the axial ratio of the ellipse is greater than 1.3, the relative bandwidth of the antennas is almost unchanged. Considering the miniaturization requirement of the antennas, the axial ratio of the ellipse is set as 1.3. (See the notes in Supplementary Fig. 12 in the revised Supporting Information)

The material system and full-wave simulation are combined based on sheet resistance for the first time to accurately predict the performance of Ti_3C_2 -10 μm antennas (Supplementary Fig. 12b). In comparison with conductivity, the sheet resistance is more accurate for full-wave simulation in Ti_3C_2 microwave performance analysis. The actual bandwidth of Ti_3C_2 -10 μm antennas is consistent with the simulated bandwidth. Thus, the design rationality of antenna size is verified (See the notes in Supplementary Fig. 12 in the revised Supporting Information).

4. The authors presented antennas with a gain below 4 dB and radiation efficiency below 80%. What can be done to increase the gain and the radiation efficiency of the antenna?

Answer: We sincerely appreciate the reviewer for the helpful suggestion. Since the proposed antenna is an omnidirectional antenna for multipath communication applications, the maximum gain of 3.2 dBi for Ti_3C_2 -10 μm antenna is already within an acceptable range due to the influence of conductor loss, dielectric loss and equivalent electrical length of the antenna.

For a certain of omnidirectionality, the gain of the antenna can be increased through the following two strategies. (i) In terms of materials, the conductivity of Ti_3C_2 component can be further improved through increasing the large single-layer ratio, narrowing flake size distribution, and reducing layer spacing between the Ti_3C_2

nanosheets (*Nat. Commun.* **13**, 3223 (2022)). (ii) In terms of antenna design and measurement, we can use alternative substrate with lower dielectric loss (i.e., Rogers 5880), increase antenna orientation and combine multiple antennas into an antenna array (Kraus, J. D. & Marhefka, R. J. *Antennas: For All Applications, Third Edition*).

The according changes have been made in our revised manuscript: “It can be further increased through improving the conductivity of Ti_3C_2 component, using alternative substrate with lower dielectric loss, increasing antenna orientation, or combining multiple antennas into an antenna array.” (Page 11, Line 12~15)

For a certain of omnidirectionality, the radiation efficiency of the antenna can be increased through the following two strategies. (i) In terms of materials, the conductivity of Ti_3C_2 component can be further improved through increasing the size of individual nanosheets and reducing layer spacing between the Ti_3C_2 nanosheets. (2) In terms of antenna design and testing, we can use alternative substrate with lower dielectric loss (i.e., Rogers 5880) (Kraus, J. D. & Marhefka, R. J. *Antennas: For All Applications, Third Edition*).

The according changes have been made in our revised manuscript: “which may be further increased through improving the conductivity of Ti_3C_2 layer or adopting suitable substrates with lower dielectric loss.” (Page 11, line 16~18)

5. In “*Adv. Mater.*, vol. 33, no. 1, pp. 1–7, 2021, doi: 10.1002/adma.202003225.” the authors purposed antennas with higher radiation efficiency a. What would be the advantage of your work to theirs?

Answer: We sincerely appreciate the reviewer for the wise suggestion. The work (*Adv. Mater.*, **33**, 1, 1-7 (2022)) does show a unique advantage of higher radiation efficiency. However, its bandwidth is very narrow and can only radiate in the upper half of the substrate. In order to satisfy different frequency requirements, the microstrip antennas of different sizes need to be designed. Moreover, the construction of flexible Ti_3C_2 antenna involves in the utilization of polyethylene terephthalate and double-sided tape between Ti_3C_2 layer and commercial circuit boards, which causes the complicated manufacture process as well as hinders the direct conformal integration with flexible

electronics and chips. In comparison, the ultrawideband Ti₃C₂ monopole antenna in our work has many specific advantages as follows.

First, the most intuitive advantage is that it can cover many commercial frequency bands at the same time, such as WiFi, Bluetooth, WCDMA, 4G TD-LTE, 5G (n41, n78). According to Shannon's equation,

$$C = B \times \log_2 \left(1 + \frac{S}{N} \right) (\text{bit} / \text{s})$$

when the signal-to-noise ratio (S/N) is the same, a larger bandwidth (B) can provide a larger data transmission rate (C). The specific real-time application of the flexible ultrawideband Ti₃C₂ monopole antennas in this work is promising in various scenarios including human-computer interaction fields (i.e., smart medical treatment, individual combat, etc), IoT (i.e., real-time sensing, identity recognition, near-field communication, etc), mobile communication systems, large data transfer, video calls, multi-person online conferences, and information exchange of large data volumes (Page 14, line 2~7). The ultrawideband Ti₃C₂ monopole antenna is an omnidirectional antenna, which has a wider radiation angle and shows the advantage in a general mobile communication environment. The bandwidth and center frequency of Ti₃C₂ antenna can be well maintained and the gain differences fluctuate within ± 0.2 dBi at the low frequency range after the bent antenna returns to the flat state (Page 2, line 10~12). The excellent cyclic bending stability ensures the fluent real-time wireless transmission for movie trailers in bending states, which is also the first demo instance of Ti₃C₂ antenna in recently reported works (Page 15, line 1~3). The transmission effect can also be achieved when the antennas are in the non-line of sight or at different orientation angles (Page 13, line 17~20).

Second, our work first exploits flexible ultrawideband Ti₃C₂ monopole antenna prepared through the progressive direct extrusion printing technology. The polydopamine, as "molecular glue" nano-binder, acts as a secondary platform to improve the interfacial adhesion interactions between dielectric substrate and printed Ti₃C₂ film, contributing the conformal integrated Ti₃C₂ antennas with the high spatial uniformity and mechanical flexibility (Page 6, line 7~10).

6. Does humidity or heat affect the antenna's performance over time?

Answer: We sincerely appreciate the reviewer for the insightful suggestion. The humidity and heat effect on the antenna performance over time has been validated in an isolated custom-made chamber.

The humidity effect on the antenna performance has been evaluated through the implementation of Ti_3C_2 antenna as radiating and sensing element while the antenna sensor is connected to the vector network analyzer (Supplementary Fig. S30a). The antenna sensor is positioned inside the sealed custom-made chamber, and the humidity is pumped by the humidifier. Resonant peak is used as the initial frequency to measure the resonant frequency shift during humidity sensing measurement (Supplementary Fig. S30b,c). When the humidity is less than 60%, the bandwidth of the antenna remains unchanged, and the amplitude of the resonance point of S_{11} increases sharply. When the humidity is greater than 60%, the bandwidth of the antenna becomes smaller and the S_{11} curve changes slowly. As the humidity further increases to 90%, the amplitude of the S_{11} curve of the antenna correspondingly increases, and the resonance point takes a left-shift. The shift in the resonant frequency of the antenna sensor can be attributed to two reasons. (i) The penetration of water molecules between the Ti_3C_2 nanosheets widens the interlayer spacing, which consequently results in the increased resistivity of the Ti_3C_2 membrane (*Nat. Commun.* **13**, 3223 (2022); *Adv. Mater. Interfaces* 2102411 (2022); *ACS Appl. Nano Mater.* **2**, 948 (2019)). The surface loss of Ti_3C_2 antenna increases, which further results in the decreased Q value (*IEEE Trans. Microw. Theory Tech.* **23**, 522-526 (1975)). (ii) Due to the humidity change, the dielectric constant of the ambient environment changes, which results in the serious mismatch between the antenna and the free space impedance. (See the notes in Supplementary Fig. 30 in the revised Supporting Information)

Supplementary Fig. 30 | The humidity effect on the antenna performance. a, Humidity sensing experimental setup. **b-c,** The antenna sensor response to different concentrations of humidity.

The heat effect on the antenna performance has been measured through the implementation of Ti_3C_2 antenna as radiating and sensing element while the antenna sensor is connected to the vector network analyzer (Supplementary Fig. S31a). The antenna is irradiated by an infrared lamp (317 mW cm^{-2}) as a heat source, and the thermal imager is used to monitor the real-time temperature (Supplementary Fig. S31b). As the temperature of the antenna rises, the amplitude of S_{11} curve decreases and the resonance point takes a right-shift (Supplementary Fig. S31c,d). It successfully demonstrates the potentials of the proposed Ti_3C_2 antenna sensor in detecting the temperature of 30-75 °C. The shift in the resonant frequency of the antenna sensor can be attributed to two reasons. (i) Ti_3C_2 possesses excellent photothermal conversion capability. Under near-infrared radiation, the enhanced photothermal effect of Ti_3C_2 results in the increased temperature, which increases the electrical conductivity of Ti_3C_2 film (*Sens. Actuat. B Chem.* **352**, 131059 (2022); *Compos. Part B* **217**, 108902 (2021); *Chem. Eng. J.* **430**, 132605 (2022)). Thus, the surface loss of the antenna decreases, and Q value increases, which results in the downward shift of the resonant points (*IEEE Trans. Microw. Theory Tech.* **23**, 522-526 (1975)). (ii) If Ti_3C_2 film is modeled as a parallel lossy resonant circuit (RLC), the increased conductivity will lead to the decreased resistive component, which reduces the impedance mismatching losses among the Ti_3C_2 membrane, the antenna feed structure, and the air medium, thus causing the right-shift of the resonant frequency (*ACS Appl. Mater. Interfaces* **14**, 6203

(2022); *Adv. Mater. Interfaces* 2102411 (2022)). (See the notes in Supplementary Fig. 31 in the revised Supporting Information)

Supplementary Fig. 31 | The heat effect on the antenna performance. **a**, The heat effect experimental setup. **b**, Recording picture of thermal imager. **c-d**, The antenna sensor response to different temperatures.

7. Can the proposed antenna be used for sensing applications which are a top trend in IoT studies? If yes can the antenna itself act as a sensor? If yes, what can be the sensing application for the proposed antenna?

Answer: We sincerely appreciate the reviewer for the constructive suggestion. The proposed antenna can be used for sensing applications in cutting-edge IoT studies. The antenna itself can act as humidity sensor and temperature sensor. In our revised manuscript, the proposed antenna has been used in the sensing application for humidity and heat effect.

The humidity effect on the antenna performance has been evaluated through the implementation of Ti_3C_2 antenna as radiating and sensing element while the antenna sensor is connected to the vector network analyzer (Supplementary Fig. S30a). The antenna sensor is positioned inside the sealed custom-made chamber, and the humidity is pumped by the humidifier. Resonant peak is used as the initial frequency to measure the resonant frequency shift during humidity sensing measurement (Supplementary Fig. S30b,c). When the humidity is less than 60%, the bandwidth of the antenna remains unchanged, and the amplitude of the resonance point of S_{11} increases sharply. When the humidity is greater than 60%, the bandwidth of the antenna becomes smaller and the S_{11} curve changes slowly. As the humidity further increases to 90%, the amplitude of the S_{11} curve of the antenna correspondingly increases, and the resonance point takes a left-shift. The shift in the resonant frequency of the antenna sensor can be attributed to two reasons. (i) The penetration of water molecules between the Ti_3C_2 nanosheets widens the interlayer spacing, which consequently results in the increased resistivity of the Ti_3C_2 membrane (*Nat. Commun.* **13**, 3223 (2022); *Adv. Mater. Interfaces* 2102411 (2022); *ACS Appl. Nano Mater.* **2**, 948 (2019)). The surface loss of Ti_3C_2 antenna increases, which further results in the decreased Q value (*IEEE Trans. Microw. Theory Tech.* **23**, 522-526 (1975)). (ii) Due to the humidity change, the dielectric constant of the ambient environment changes, which results in the serious mismatch between the antenna and the free space impedance. (See the notes in Supplementary Fig. 30 in the revised Supporting Information)

Supplementary Fig. 30 | The humidity effect on the antenna performance. a, Humidity sensing experimental setup. **b-c,** The antenna sensor response to different concentrations of humidity.

The heat effect on the antenna performance has been measured through the implementation of Ti₃C₂ antennas as radiating and sensing elements while the antenna sensor is connected to the vector network analyzer (Supplementary Fig. S31a). The antenna is irradiated by an infrared lamp (317 mW cm⁻²) as a heat source, and the thermal imager is used to monitor the real-time temperature (Supplementary Fig. S31b). As the temperature of the antenna rises, the amplitude of S₁₁ curve decreases and the resonance point takes a right-shift (Supplementary Fig. S31c,d). It successfully demonstrates the potentials of the proposed Ti₃C₂ antenna sensor in detecting the temperature of 30-75 °C. The shift in the resonant frequency of the antenna sensor can be attributed to two reasons. (i) Ti₃C₂ possesses excellent photothermal conversion capability. Under near-infrared radiation, the enhanced photothermal effect of Ti₃C₂ results in the increased temperature, which increases the electrical conductivity of Ti₃C₂ film (*Sensor Actuat. B: Chem.* **352**, 131059 (2022); *Compos. Part B* **217**, 108902 (2021); *Chem. Eng. J.* **430**, 132605 (2022)). Thus, the surface loss of the antenna decreases, and *Q* value increases, which results in the downward shift of the resonant points (*IEEE Trans. Microw. Theory Tech.* **23**, 522-526 (1975)). (ii) If Ti₃C₂ film is modeled as a parallel lossy resonant circuit, the increased conductivity will lead to the decreased resistive component, which reduces the impedance mismatching losses among the Ti₃C₂ membrane, the antenna feed structure, and the air medium, thus causing the right-shift of the resonant frequency (*ACS Appl. Mater. Interfaces* **14**, 6203 (2022); *Adv. Mater. Interfaces* 2102411 (2022)). (See the notes in Supplementary Fig. 31 in the revised Supporting Information)

Supplementary Fig. 31 | The heat effect on the antenna performance. a, The heat effect experimental setup. b, Recording picture of thermal imager. c-d, The antenna sensor response to different temperatures.

8. Figure 4l shows the transmission of data between two MXene-based antennas. Can this communication also happen in the long-range in the range of meters? Can the authors present the results of this communication? Also, I am interested to see the communication between the antenna where one antenna transmits a signal generated by a signal generator and the receiving antenna reveals the response in a spectrum analyzer.

Answer: We sincerely appreciate the reviewer for the insightful suggestion.

Fig. 4l shows the data transmission between two Ti_3C_2 antennas. The real-time communication can also happen in the long-range distance of 1-5 m (Supplementary Fig. 28 and Supplementary Video 7). In the experimental setup, the left antenna is the transmitting antenna, and the right antenna is the receiving antenna. When the distance

is less than 4 m, the transmission effect of the antenna is very fluent, the points on the planisphere are very concentrated, the error rate is almost 0, and the video transmission is very stable (Supplementary Fig. 28a-d). When the distance is extended to 5 m, the video transmission becomes slightly unstable, but the overall transmission effect is still acceptable (Supplementary Fig. 28e). (See the notes in Supplementary Fig. 28 in the revised Supporting Information)

Supplementary Fig. 28 | Real-time video transmission when two Ti_3C_2 antennas are in the range of meters. a, 1 m, b, 2 m, c, 3 m, d, 4 m, e, 5 m.

The communication between the antennas where one antenna transmits a signal generated by the signal generator and the receiving antenna reveals the response has been revealed in a spectrum analyzer (Supplementary Fig. 29). The transmitting and receiving antennas are both Ti_3C_2 antennas fabricated by us. The distance between the two antennas is 2 m. The signal generator produces 2-4 GHz signals with 0.2 GHz step and the signal power is 20 dBm. The receiving power of the antenna on the spectrum analyzer can be further calculated according to the free space path loss equation (S2) and Friis equation (S3) (Proc. Ire, **34**, 254-256 (1946)).

$$L_{\text{path}} \text{ (dB)} = 32.45 + 20 \lg r \text{ (km)} + 20 \lg f \text{ (MHz)} \quad (\text{S2})$$

where r represents the distance between the two antennas, and f represents the operating frequency of two antennas.

$$P_r \text{ (dB)} = [P_t - L_{\text{path}} + 10\lg G_r + 10\lg G_t] \text{ (dB)} \quad (\text{S3})$$

where P_r represents the receiving power, P_t represents the transmitting power, $10\lg G_r$ represents the gain of the receiving antenna, $10\lg G_t$ represents the gain of the transmitting antenna.

For example, the free space path loss is 48.01 dB at 3.0 GHz, and the gains of the receiving antenna and the transmitting antenna are both 2 dBi. Thus, the receiving power on the spectrum analyzer can be calculated as -24.01 dBm. In addition, the transmission line and the connector will have about 1.5 dB loss in the actual measurement, so the actual receiving power should be -25.51 dBm. In addition, there are some noises of measurement results on Industrial Scientific Medical band caused by wireless systems nearby. (See the notes in Supplementary Fig. 29 in the revised Supporting Information)

Supplementary Fig. 29 | The communication response in a spectrum analyzer. a, The experimental setup. **b,** Transmission measurement results.

9. Are the communication measurement results repeatable over time?

Answer: We sincerely appreciate the reviewer for the helpful suggestion. The communication measurement results are repeatable for at least one month

(Supplementary Fig. 14). The long-range stability will continue to be tested in our future work.

Supplementary Fig. 14 | The reflection coefficient S_{11} repeatable for one month.

10. Can the antennas communicate when they are not in the line of sight or have different orientation angles?

Answer: We are very grateful for the reviewer's wise suggestion.

The antenna communication can be achieved when they are in the non-line of sight (Supplementary Fig. 26 and Supplementary Video 5). The transmitting antenna is set in left side, and the receiving antenna is set in right side (Supplementary Fig. 26a). When they are in the line of sight, the video transmission is very stable. When the oil painting is used to separate the two antennas, the video transmission effect is not affected and remains very stable (Supplementary Fig. 26b,c). When the two antennas are separated by a metal plate, the video transmission suddenly deteriorates for a short time (Supplementary Fig. 26d,e). However, due to the multipath effect in the mobile communication environment, the video can be transmitted after a few seconds. After removing the metal plate, the video transmission becomes stable again (Supplementary Fig. 26f). (See the notes in Supplementary Fig. 26 in the revised Supporting Information)

Supplementary Fig. 26 | Real-time video transmission when two Ti_3C_2 antennas are in the non-line of sight. (a-c) Separated by oil painting, (d-f) Separated by metal plate.

The antennas can also communicate at different orientation angles (Supplementary Fig. 27 and Supplementary Video 6). When the side of one antenna is versus the front of another antenna (Supplementary Fig. 27a) or the back of one antenna is versus the front of another antenna (Supplementary Fig. 27b), the video transmission is very stable. When the transverse direction of one antenna is versus the forward direction of another antenna (Supplementary Fig. 27c), the video transmission becomes slightly worse. This phenomenon is caused by the different polarization directions of the two antennas. When the head of an antenna is versus the side of another antenna (Supplementary Fig. 27d), the video transmission further deteriorates. This phenomenon is caused by the fact that the polarization direction of the two antennas is different and the maximum

gain direction is not aligned. When the head of an antenna is versus the head of another antenna (Supplementary Fig. 27e), the video transmission almost stops working. The reason for this phenomenon is that both antennas are aligned in the direction of minimum gain.

Supplementary Fig. 27 | Real-time video transmission when two Ti₃C₂ antennas have different orientation angles. **a**, The side of one antenna versus the front of another antenna. **b**, The back of one antenna versus the front of another antenna. **c**, The transverse direction of one antenna versus the forward direction of another antenna. **d**, The head of an antenna versus the side of another antenna. **e**, The head of an antenna versus the head of another antenna.

11. *The authors are encouraged to present a table for the thickness of MXene, the conductivity and the skin depth of the MXene which they used.*

Answer: We are very grateful for the reviewer’s wise suggestion. The table for the thickness, conductivity, and skin depth of Ti₃C₂ has been presented in Supplementary Table 2 in the revised Supporting Information.

Supplementary Table 2. The thickness, conductivity, and skin depth (at 2.4 GHz) of Ti₃C₂ antenna.

Thickness / μm	Conductivity / S cm^{-1}	Skin depth / μm
3	7765.84	11.66
5.5	7221.39	12.08
10	4474.07	15.36

The skin depth is calculated by the Equation (S4),

$$\delta = \sqrt{1/\pi\sigma\mu f} \quad (\text{S4})$$

where σ , μ , and f are conductivity, permeability, and frequency, respectively (*Sci. Adv.* **4**, eaau0920 (2018)). (See the notes in Supplementary Table 2 in the revised Supporting Information).

12. *The authors mentioned for different bending angles that “The bandwidth and center frequency can be well maintained and the gain differences fluctuate within ± 1.0 dBi.”. This gain error margin is high for an antenna with a gain of 4 dB.*

Answer: We are very grateful for the reviewer's wise suggestion. In this work, the gain error margin fluctuates within ± 1.0 dBi in working frequency band of 1.7-4.0 GHz after 1000 bending cycles. The maximum gain difference is ± 1.0 dBi at a certain frequency point. The main reasons for the high gain error margin are as follows. (i) In our work, the antenna is proposed for omnidirectional radiation in multiple communication environments in which the gain error margin is acceptable for an antenna with a gain of 4 dBi. (ii) There is no large metal ground on the back of the Ti_3C_2 antenna. It makes the antenna gain very sensitive to small deformations of the dielectric substrate. (iii) The working bandwidth of ultrawideband Ti_3C_2 monopole antenna is from 1.7 GHz to 4.0 GHz. There are many measurement values that may introduce some singularities after 1000 bending cycles for the Ti_3C_2 antennas with different thicknesses. Therefore, it is reasonable that there are some special fluctuated points in the range of $0 - \pm 1.0$ dBi at a certain frequency point. Moreover, the gain difference fluctuates within ± 0.2 dBi at the low frequency range for Ti_3C_2 -3 μm and Ti_3C_2 -5.5 μm antenna, which is comparable to the previously reported work (*Adv. Mater.* 2003225 (2020)) (Page 12, line 13~15).

In this work, the gain error margin for the ultrawideband Ti_3C_2 monopole antenna is rational and acceptable. We will further reduce the gain difference through process optimization in our future work.

Finally, we would like to thank the reviewer again for these valuable comments and for the thoughtful and careful review towards improving our manuscript.

REVIEWERS' COMMENTS

Reviewer #1 (Remarks to the Author):

Good work. The authors have addressed all the queries raised by the reviewer. Acceptance of the script is recommended.

Reviewer #2 (Remarks to the Author):

I really appreciate the additional work made for improving the paper. I recommend the publication to this journal

Reviewer #3 (Remarks to the Author):

My comments have been addressed. no further comments or concerns.

Point-by-Point Response to Referees

Reviewer #1:

Good work. The authors have addressed all the queries raised by the reviewer. Acceptance of the script is recommended.

Response: We sincerely appreciate the positive comment from reviewer.

Reviewer #2:

I really appreciate the additional work made for improving the paper. I recommend the publication to this journal.

Response: We are sincerely grateful for your positive comments on our work.

Reviewer #3:

My comments have been addressed. no further comments or concerns.

Response: We appreciate the recognition to our work from reviewer.